# Fluorescent Probes for Live Cell Thiol Detection

**DOI:** 10.3390/molecules26123575

**Published:** 2021-06-11

**Authors:** Shenggang Wang, Yue Huang, Xiangming Guan

**Affiliations:** Department of Pharmaceutical Sciences, College of Pharmacy and Allied Health Professions, South Dakota State University, Box 2202C, Brookings, SD 57007, USA; Shenggang.wang@fda.hhs.gov (S.W.); Yue.huang@sdstate.edu (Y.H.)

**Keywords:** live cell thiol fluorescence imaging, cellular thiols, subcellular thiols, non-protein thiols, protein thiols, mitochondrial thiols, Lysosomal thiols, glutathione, cysteine, homocysteine, thiol specific agents, thiol-sulfide exchange reaction, thiol-disulfide exchange reaction, Michael addition reaction

## Abstract

Thiols play vital and irreplaceable roles in the biological system. Abnormality of thiol levels has been linked with various diseases and biological disorders. Thiols are known to distribute unevenly and change dynamically in the biological system. Methods that can determine thiols’ concentration and distribution in live cells are in high demand. In the last two decades, fluorescent probes have emerged as a powerful tool for achieving that goal for the simplicity, high sensitivity, and capability of visualizing the analytes in live cells in a non-invasive way. They also enable the determination of intracellular distribution and dynamitic movement of thiols in the intact native environments. This review focuses on some of the major strategies/mechanisms being used for detecting GSH, Cys/Hcy, and other thiols in live cells via fluorescent probes, and how they are applied at the cellular and subcellular levels. The sensing mechanisms (for GSH and Cys/Hcy) and bio-applications of the probes are illustrated followed by a summary of probes for selectively detecting cellular and subcellular thiols.

## 1. Introduction

Thiols, compounds that contain a sulfhydryl group (-SH), in the biological system are referred as biological thiols or biothiols [1,2]. Biothiols can be categorized into two classes: protein thiols (PSH) and non-protein thiols (NPSH). NPSH are mainly small molecule thiols including cysteine (Cys), homocysteine (Hcy), and glutathione (GSH), while PSH are primarily the cysteine residues in proteins [3]. Thiols play vital and irreplaceable roles in various cellular functions in the biological system [4] thanks to their unique chemical properties of strong nucleophilicity, reductivity, and chelating ability for metals. These roles include detoxification to remove reactive electrophiles, as an antioxidant to remove reactive oxygen species (ROS), and the ability of metal chelation. For example, GSH is the key compound in glutathione conjugation reaction—a phase II drug metabolism in the body to remove reactive electrophiles which are toxic. GSH is also the most abundant intracellular NPSH and serves as the most important antioxidant to prevent damages caused by ROS via termination of ROS. In addition, thiols are also involved in many other aspects of cellular functions, such as being part of enzyme active sites, being involved in signal transduction, being involved in cell division, etc. Abnormality of thiol levels has been linked with many diseases and biological disorders, such as cardiovascular diseases, cancer, Alzheimer’s disease and etc [5].

Thiols are distributed heterogeneously and their levels change dynamically in the biological system. Subcellular organelles such as nucleus, mitochondria, lysosomes, endoplasmic reticulum (ER), Golgi apparatus, cell surfaces, etc., are important structures that play critical roles in the normal function of cells. Disruption of the homeostasis of these structures can interrupt the normal function of cells resulting in many disorders and diseases [6,7]. Thiols distribute unevenly in these subcellular organelles [8]. Their levels in different subcellular organelles can serve as an indicator to reflect the status of the subcellular organelles. Thus, tools or methods that can map thiol distribution and monitor their status changes will provide valuable information to understand the relationship between thiols and cells’ functions/dysfunctions, and help guide the correlation/treatment of these dysfunctions and their related diseases [9].

Many analytical methods have been developed for detecting thiols in the biological system. These methods include high performance liquid chromatography (HPLC/UV) [10,11,12], mass spectrometry (MS), HPLC-MS [13], colorimetric assay [14], enzyme assays, capillary electrophoresis, and gel electrophoresis [15,16,17,18]. Most of these methods need a complicated and lengthy sample preparation step and involve a homogenization process to break cells or tissues before an analytical method can be applied. Due to chemical instability of thiols, the sample process step quite often can introduce potential artifacts and cause a potential loss of the analyte.

In the past two decades, one of the most popular approaches being developed to address analyte loss during cell/tissue homogenization is to analyze the analyte in live cells using the fluorescence imagining technique via the use of a fluorescent probe [19]. Fluorescence imaging outstands itself by its simplicity, high sensitivity, and capability of visualizing the analytes in live cells in a non-invasive way [1]. The non-invasive fluorescence imaging provides the advantage of visualizing an analyte without breaking the cell [20,21,22,23,24,25,26,27], enabling the determination of intracellular distribution and dynamitic movement of an analyte in its intact native environments, and reveal information that cannot be revealed after cells/tissues are homogenized.

Significant efforts have been made for the detection of thiols in the biological system via fluorescence imaging by developing a large number of fluorescent probes for thiols. These probes include BODIPY [28,29], rhodamine [30,31], monochlorobimane [32], fluorescein [33,34], tetraphenylethylene [35], coumarin [36,37], curcumin [38], mercury orange [32], rosamine-based [39], naphthalimide-based [40], polymethine [41], benzofurazan [42], and pyrene [43] (Figure 1). The optical spectra of these fluorescence probes cover visible fluorescence, near infra-red (NIR) fluorescence, single-photon or two-photon excitation fluorescence, or aggregation-induced emission (AIE) fluorescence. Some of these probes are developed as an “off-on” agents in a single channel which themselves are non-fluorescent due to a variety of quenching mechanisms (e.g., FRET, PET, and ICT) but can be turned on to a fluorescent product by thiols. The others are “ratiometric” probes utilizing dual-channel emission which themselves exhibit fluorescence but can be switched to a totally different fluorescence upon reaction with thiols and the thiol’s level can be reflected by calculating the ratio of the two fluorescence intensities. Compared with “off-on” probes which measure the absolute intensity at only one wavelength, ratiometric fluorescent probes are believed to be more accurate since the use of two different emission wavelength can provide a built-in correction [44].

Among the reported probes, majority of them were designed to detect thiols from the whole cell despite the fact that thiols are distributed unevenly inside cells. Only a few probes were for the detection of mitochondrial thiols while fewer for lysosomes and endoplasmic reticulum (ER). Finding a method to detect thiols in subcellular organelles remains to be a challenge.

For a fluorescence imaging agent to image thiols in live cells, the agent needs to be cell membrane permeable, capable of turning thiols selectively, ideally specifically, into fluorescence strong enough for detection, and noncytotoxic or at least noncytotoxic during the imaging experiment. For a subcellular thiol imaging agent, the agent needs to be cell membrane permeable, capable of turning thiols selectively, ideally specifically, in the targeted subcellular organelle into fluorescence strong enough for detection, and noncytotoxic or at least noncytotoxic during the imaging experiment. For a cellular or subcellular thiol quantifying fluorescence imaging agent, it requires the reagent to quantitatively turn thiols into fluorescence strong enough for detection within a reasonable time in addition to the requirements for a cellular or subcellular thiol fluorescence imagining agent.

This review will provide an overview of the fluorescent probes that are developed for detecting thiols in live cells. These probes will be mainly classified into two categories: i. probes used for detection of GSH or total thiols; ii. probes used to detect Cys/Hcy. Intracellularly, GSH is much more abundant (1–10 mM) than Cys (30–200 μM) and Hcy (5–12 μM) [45]. The detection of intracellular GSH is less likely to be interfered by Cys/Hcy. As a result, GSH is often used to reflect the total NPSH in cells. Therefore, the probes claimed for GSH or total NPSH are summarized as one category in this review although some of the probes were claimed to be selective for GSH. In contrast, the detection of Cys/Hcy can be interfered by GSH. For that reason, the probes designed for distinguishing Cys/Hcy over GSH are summarized into a different category.

The purpose of this review is not trying to be inclusive in terms of thiol fluorescence probes since they have been covered by other comprehensive reviews [2,46,47,48,49]. Rather, this review focuses on summarizing some of the major strategies or mechanisms that are used for detecting GSH, Cys/Hcy, and how they are applied for the detection of cellular or subcellular thiols. The fluorophores being used, sensing mechanisms (for GSH or Cys/Hcy) and bio-applications of the probes will be illustrated followed by a summary of probes for selectively detecting cellular and subcellular thiols.

## 2. Detection of Thiols in Cells

### 2.1. NPSH, GSH, Cys, and Hcy

NPSH are thiols other than PSH. They include GSH, Cys, Hcy, and other small molecule thiols. Among them, GSH is the most abundant thiol with a concentration of millimolar in the biological system. GSH serves as the most important antioxidant to prevent damages caused by ROS [50,51] via terminating ROS with itself being converted to its oxidized form glutathione disulfide (GSSG). GSSG is then converted back to GSH by glutathione reductase. Cells maintain a normal ratio (>100) of GSH/GSSG through the GSH/GSSG cycle [52,53]. The abnormal level of GSH has been associated with various diseases such as cancer, AIDS, growth delay, neurodegenerative disease, and cardiovascular disease [54,55,56,57]. Thus, determination of GSH level in live cells is of particular interest.

Numerous fluorescent probes have been developed for the detection of intracellular GSH or thiols in live cells based on different mechanisms. Most of these methods were designed based on the high nucleophilicity of thiols [58,59].

Distinguishing GSH from Cys and Hcy remains a challenge due to the structure and reactivity similarities GSH shared with Cys/Hcy [60,61,62]. The majority of the thiol probes being developed are not able to distinguish GSH from other thiols including Cys/Hcy, since most of them undergo similar reactions and similar emission changes upon reaction with a thiol. However, as mentioned earlier Cys or Hcy is less likely to interfere with the detection of GSH since intracellular GSH level (1–10 mM) is much higher than Cys (30–200 μM), and Hcy (5–15 μM). Although some of the probes have been reported to selectively detect GSH and some reported for detecting NPSH, all these methods are included in the category of GSH and total thiols. This part will focus on the main strategies/mechanisms that have been developed for the detection of GSH and total thiols. The strategies/mechanisms included in these methods are: A. Michael addition reaction with α, β -unsaturated carbonyl derivatives [63,64]; B. Nucleophilic substitution with substrates bearing a leaving group (e.g., halogen, ether, or thioether) [29,65,66]; C. Cleavage of sulfonamide or sulfonate ester [67]; D. Disulfide–thiol exchange reaction [68]; E. others [30,46] (Figure 2). It is noted that although reaction with a thiol is preferred in all these reactions, other nucleophiles such as -OH, or -NH can also be involved in these reactions. The involvement of other nucleophiles in the reaction will affect thiol imaging especially quantitative thiol imaging. Nevertheless, thiol–disulfide exchange reaction (D) and thiol–sulfide exchange reaction (B) have been demonstrated to be thiol specific. Thiol specific fluorescence imaging agents based on the thiol-sulfide reaction have been developed and used successfully to image and quantify cellular thiols and subcellular thiols in live cells (refer to Section 2.1.2, Section 3.1.2 and Section 3.3.2)

#### 2.1.1. Detection of NPSH via a Michael Addition Reaction

Michael addition is one of the most commonly used strategies for detection of thiols. Michael addition reaction has a high selectivity for thiols though it can also occur with other nucleophiles such as -OH, NH, or -COOH. Xiaoqiang Chen et al. [34] published their work on the development of a thiol imaging tool bearing fluorescein as the fluorophore (**Probe 1**). As shown in Figure 3, the process of turning on the fluorescence starts from a 1,4-addition of thiols to the α, β-unsaturated ketone in the probe. Subsequent spiro ring opening of fluorescein lights up the fluorescence (Ex. 485 nm, Em. 520 nm) which provides fluorescence enhancement by 61-fold compared with probe itself. The probe exhibits a similar fluorescence enhancement towards Cys, Hcy, and GSH. With fluorescein as a standard, the quantum yields of the thiols adducts with probe were determined to be 0.65, 0.91, and 0.47 for GSH, Cys, and Hcy, respectively. The probe is featured with a rapid response and high sensitivity with a detection limit of 10^−7^–10^−8^ M. Further, the probe was proved to be capable of detecting the thiol status change in live P19 cells. More importantly, the probe is also the first reagent being used to monitor the thiols status in zebrafish. A significant fluorescence intensity decrease was observed when the cells or zebrafish were treated with a thiol trapping reagent, N-methylmaleimide (NMM) (40–50 μM).

In 2011, Gun-Joong Kim et al. [36] reported a Michael addition reaction based fluorescent probe (**Probe 2**) for ratiometric imaging of cellular GSH. As illustrated in Figure 4, the probe has three parts, a coumarin moiety serving as the signaling fluorophore, a thiol reactive enone linker, and an *O*-hydroxyl group as an activation unit through a resonance-assisted hydrogen bond [69,70]. Once a reaction occurs between thiol (2-mercaptoethanol as a representative agent) and the Michael acceptor structure of the probe, the fluorescence emission band will shift from 553 to 466 nm (Ex. 420 nm). The ratio of the fluorescence intensity under these wavelength (F_466 nm_/F_553 nm_) changes ratiometrically upon reaction with thiol such as Hcy, Cys, and GSH. The probe was successfully applied for monitoring cellular thiol status in live cells. The fluorescence ratio (F_blue 410 nm–460 nm_/F_green 490–540 nm_) increased significantly when the cells were pretreated with R-lipoic acid, an enhancer of GSH. In contrast, a remarkable ratio decrease was observed when cells were pretreated with NEM, a scavenger of GSH.

It is known that maleimide can react selectively toward thiols than other nucleophiles via a Michael addition reaction. In the meantime, maleimide can also serve as a fluorescence quencher in their conjugated form. Thus, maleimide has been widely utilized by many groups to build the thiols selective probes. For instance, in 2017, Hai Shu et al. [71] reported a rhodamine B-based probe (L1, **Probe 3**) with maleimide serving as a thiol recognition unit. Upon reaction with GSH, the probe can be turned on from non-fluorescent to a strong fluorescent thiol-rhodamine adduct (586 nm) due to a Michael addition reaction followed by a spirolactam ring opening (Figure 5). A more than 200-fold increase of fluorescence intensity was observed when excessive GSH was added. In addition, the fluorescence intensity was found to be linearly increased with an increase in GSH (2–26 μM) with a detection limit of 0.219 μM. The probe shows a high selectivity toward GSH than other thiols. Further, cellular experiments demonstrated that L1 can be used to monitor intracellular GSH in HepG-2 and HUVEC cells. Another similar approach was reported by Tao Liu et al. [72] in 2016 for selective detection of thiols in live cells with a high sensitivity (**Probe 4**). The detection limits were determined to be 0.085, 0.12, and 0.13 μM for GSH, Hcy, and Cys, respectively. Similarly, the probe was also successfully applied in the determination of intracellular GSH in live HepG2 cells.

In contrast to traditional fluorescent methods, reagents which can detect a thiol in a reversible manner was developed. In 2015, Xiqian Jiang et al. [73] reported a reversible and ratiometric fluorescent probe-ThiolQuant Green (TQ Green, **Probe 5**) for quantitative imaging of GSH. The probe is composed of a 7-amino coumarin group as the fluorophore and an aromatic structure to extend the absorption wavelength of coumarin. These two parts are linked together by a thiol reactive group, Michael acceptor. The probe’s reaction with GSH is reversible when GSH is depleted. TQ Green (Ex. 488 nm, Em. 590 nm) is turned into to TQ Green-GSH (Ex. 405 nm, Em. 463 nm) upon reaction with GSH (Figure 6). The ratio of signal intensities (absorbance or fluorescence) between TQ Green-GSH and TQ Green is proportional to the GSH concentration enabling the ratiometric detection of GSH with a detection limit of 20 nM. No reaction was observed when the probe was exposed to an excessive ratio of bovine serum albumin (BSA) indicating the probe does not react with PSH. Cellular experiments with multiple cell lines, including 3T3-L1, HepG2, PANC-1, and PANC-28 cells, proved that the GSH concentration obtained from TQ Green live imaging were well correlated with the values achieved from bulk lysate measurements. The probe can well reflect the GSH level changes in live cells when the GSH level was reduced by diethylmaleate (DEM). One drawback of the method is that the reverse reaction between GSH and TQ Green is slow leading to the inability of the probe for a quick detection of a decrease of GSH in live cells.

Continuing with their work, Xiqian Jiang et al. [74] published another reversible fluorescent probe named as RealThiol (RT, **Probe 6**) in 2017 for quantitatively real-time monitor of intracellular GSH in live cells with a much higher reaction rate (50-fold faster than TQ Green). RT was developed based on a series of optimization of TQ Green to enable a quantitative real-time imaging of GSH in living cells with a much-enhanced reaction rate for both forward and reverse directions. Similar to TQ Green, RT shows ratiometric fluorescence response with a superb linear relationship to various concentrations of GSH in a range of 1–10 mM. In contrast to TQ Green which exhibits a second-order forward reaction rate of 0.15 M^−1^ s^−1^, the second-order reaction rate for RT is 7.5 M^−1^ s^−1^ which is 50 times faster than TQ Green. No significant fluorescence change was observed when RT was mixed with cell lysates suggesting the inability of RT to react with PSH. Impressively, the probe could well reflect the GSH level change when HeLa cells were treated with different concentrations of H_2_O_2_. Later, the probe was demonstrated to quantitatively monitor GSH levels when cells were consecutively treated with H_2_O_2_ (500 μM) and GSH ester (100 μM). Further, with the aid of RT, an enhanced antioxidant capability of activated neurons and dynamic GSH changes during ferroptosis were observed. RT was successfully applied for high-throughput quantification of GSH in single cells via flow cytometry.

It should be noted that a number of other reversible fluorescent probes have been developed for quantitatively monitoring GSH in real time in live cells with various fluorophores and sensing mechanisms. These probes include probe QG-1 developed by Zhixue Liu et al. [75], probe RP-1 and RP-2 by Ming Tian et al. [76,77], and a number of single-molecule localization microscopy (SMLM) applicable probes developed from Urano groups [78].

#### 2.1.2. Detection of NPSH via a Nucleophilic Aromatic Substitution (SNAr) Reaction Using Halogen, Ether, Thioether as a Leaving Group

SNAr reaction is another widely used mechanism for imaging thiols in live cells since the thiol group can easily displace a leaving group that is attached to an aromatic ring in a fluorophore by a substitution reaction to form a thioether. These leaving groups include halogen, ether, thioether, etc. To be noted here, SNAr substitution reactions have also been commonly used to discriminatively detect Cys/Hcy due to a rearrangement reaction occurring to the thioether formed in the substitution reaction. The rearrangement reaction only occurs to the thioether derived from Cys and Hcy (Figure 7).

##### Detection of NPSH via a SNAr Reaction Using Halogen as a Leaving Group

In 2012, Liya Niu and co-workers [29] reported their efforts on developing a fluorescent probe (**Probe 7**) for highly selective detection of GSH over Cys/Hcy based on monochlorinated BODIPY. As shown in Figure 7, a SNAr reaction occurred between the sulfur group (GSH, Cys, or Hcy) and the monochlorinated BODIPY resulting in a thioether. A fluorescence emission wavelength red shift (556 to ~588 nm) occurred after the reaction, and the ratio of fluorescence intensity (I_588_/I_556_) linearly increased as an increase in the concentration of GSH (0−60 μM) with the detection limit of 8.6 × 10^−8^ M, enabling the quantification of intracellular GSH in live cells. The probe was successfully applied for detecting the GSH levels’ change when the cells were pretreated with diamide [79] in HeLa cells. The probe was claimed to be the first ratiometric fluorescent sensor developed for the selective detection of GSH over Cys/Hcy.

A quick intramolecular displacement occurred with the thioether generated from Cys and Hcy to replace the thiolate with the amino group to form amino-substituted BODIPY, while the thioether generated from GSH was not able to undergo the intramolecular displacement due to the unavailability of an amino group. Since the photophysics of BODIPY derivatives is sensitive to substituents, the amino-substituted BODIPY exhibit a relative weaker and blue shifted fluorescence which enable the selective detection of GSH over Cys and Hcy, while the rapid intramolecular displacement of S to N will shift the emission maxima back to around 564 nm for Cys and Hcy, with GSH remaining at 588 nm.

##### Detection of NPSH via a SNAr Reaction Using ether as a Leaving Group

In addition to halogen, other leaving groups, such as ether or thioether are also widely used as a thiol recognition unit for building a thiol selective probe. For example, Xinghui Gao et al. [80] developed a probe (**Probe 8**) in 2015 for detection of GSH and Cys in different emission channels by connecting 7-nitrobenzofurazan with resorufin through the ether bond in which the ether bond also served as a thiols recognition unit. Reaction with thiols (GSH, or Cys) yields a free strong fluorescent resorufin (Em. 585 nm) and a thiol conjugated nitrobenzofurazan. However, the thiol conjugated notrobenzofurazn derived from Cys will undergo an intramolecular rearrangement to generate a new fluorescent product with λem = 540 nm while the thiol conjugated notrobenzofurazn derived from GSH remained unchanged (Figure 8). Measuring the fluorescence intensity from resorufin (Em. 585 nm) enabled the determination of the total thiols (GSH, and Cys/Hcy) while measuring the fluorescence intensity of both resorufin and the thiol conjugated nitrobenzofurazan provided a method to distinguish GSH from Cys. The detection limits for GSH and Cys were reported to be 0.07 μM and 0.13 μM respectively with a linear relationship between the fluorescence intensity and the concentration of GSH (1–18 µM) and Cys (3.1–90 µM, 1–40 µM), respectively. The probe was employed to detect Cys and GSH in human plasma.

Xilei Xie et al. [81] reported a probe (Res-Biot, **Probe 9**) in 2017 for selective detection of GSH based on the ether bond. In this approach, pyrimidine moiety was for the first time investigated and optimized as a unique recognition unit for thiols. The pyrimidine moiety was connected with fluorophore resorufin via an ether linkage to generate probe Res-Biot. Upon reaction with a thiol, a high fluorescent resorufin is released with a strong fluorescence at 585 nm (Figure 9). Moreover, the fluorescence intensity at 585 nm was increased proportionally to the concentrations of GSH in a range of 0−20 μM with a detection limit of 0.29 μM. The probe was successfully used to detect a decrease in intracellular GSH concentration down-regulated by an oxidative stress inducer, 12-myristate 13-acetate (PMA), or by L-buthionine sulfoximine (BSO)—an inhibitor of γ-glutamylcysteine synthetase.

##### Detection of NPSH via a SNAr Reaction Using Thioether as a Leaving Group

In 2012, Yinghong Li et al. from our group reported a thiol specific probe [42] (GUALY’s reagent, **Probe 10**) based on a symmetric benzofurazan sulfide for imaging and quantifying total thiols in live cells. The symmetric benzofurazan sulfide was constructed by combining two benzofurazan fluorophores with a thioether group (Figure 10). The probe itself is not fluorescent due to the self-quenching of the symmetric structure but turns into a fluorescent product (Ex. 430 nm, Em. 520 nm) rapidly upon reaction with thiols through a thiol specific thiol–sulfide exchange reaction. GUALY’s reagent has been successfully applied for imaging and quantifying total thiols (PSH+NPSH) in live cells [42,82].

Jing Liu et al. [83] reported a GSH discriminating fluorescent probe (**Probe 11**, Figure 11) in 2014 based on pyronin B as the fluorophore and thioether as the thiol recognition group. In the approach, A methoxythiophenol group was connected to the fluorescent pyronin moiety to serve as a fluorescence quencher via a photoinduced electron transfer (PET) process. Reaction with thiols (GSH, Cys, Hcy) through SNAr reaction generates a highly fluorescent thiol conjugated pyronin with λ_em_ for GSH-pyronin and Cys/Hcy-pyronin at 622 nm and 546 nm respectively due to the additional intramolecular rearrangements of Cys/Hcy. The probe has been successfully applied for simultaneously detection of Cys/Hcy and GSH in B16 cells.

In 2015, Lun Song et al. [84] reported two water-soluble colorimetric and turn-on fluorescent probes (STP1, 2, **Probes 12**, **13**) for selective detection of GSH using thioether as a thiol recognition group and naphthalimide as the fluorophore reporter. The probes are non-fluorescent due to a PET process from the naphthalimide electron donor to 4-nitrobenzene acceptor via the thioether bond (Figure 12). Upon reaction with GSH, a remarkable enhancement of fluorescence (~90-fold) at 487 nm was observed. A linear relationship was found between the fluorescence intensity (490 nm) and GSH concentration in the range of 0–100 μM with the detection limit of 84 nM. It should be noted that a linear relation was also found between the fluorescence intensity and thiol-containing proteins bovine serum albumin (BSA) and ovalbumin (OVA) suggesting that the probes can also imaging PSH. These probes have been demonstrated to be capable for fluorescence imaging of GSH in HeLa cells.

#### 2.1.3. Detection of NPSH via Cleavage of Sulfonamide or Sulfonate Ester

##### Detection of NPSH via Cleavage of Sulfonamide

In 2014, Masafumi Yoshida et al. [85] reported a novel cell-membrane-permeable fluorescent probe (**Probe 14**) for detection of GSH based on the cleavage of sulfonamide as the thiol responsive mechanism. As illustrated in Figure 13, a fluorophore hydroxymethylrhodamine green (HMRG) and a 2,4-dinitrobenzenesulfonyl (DNBS) moiety were combined by a sulfonamide group in the approach. The fluorescence of the probe was quenched by two fluorescence quenching mechanisms, intramolecular spirocyclization (close-ring form of HMRG) and intramolecular PET between HMRG and DNBS. Reaction with a thiol breaks the quenching by cleaving the sulfonamide bond to release a free open-ring high fluorescent HMRG with an up to 7000-fold increase in fluorescence intensity (520 nm) and the increase was proportional to the concentration of GSH around the physiological concentration. The probe was successfully applied for the detection of GSH levels’ change in live cells. Further, the probe was used to detect tumor nodules in tumor bearing mice of SHIN-3 ovarian cancer by taking advantage of the fact that GSH level in tumor tissues is higher than normal tissues.

Fluorophores with emission in red or near NIR region have the advantages of low fluorescence background, deeper penetration, and less damages to live cells or tissues [86,87]. In 2014, a cyanine-based fluorescent probe (**Probe 15**, Figure 14) for highly selective detection of GSH in cells and live mouse tissues have been reported by Jun Yin et al. [88]. The probe utilizes a 5-(dimethylamino)naphthalenesulfonamide group for highly selective detection of GSH over Cys and Hcy. In vitro cellular experiments with HeLa cells proved that the probe was capable of monitoring GSH in live cells. When the cells were pretreated with *N*-methylmaleimide (NMM, a thiol-blocking agent) followed by treatment with the probe, it showed no red fluorescence while strong fluorescence was observed when no NMM was used confirming that the probe was detecting thiols. In addition, only a minor change of fluorescence intensity was observed when Cys (100 μM) or Hcy (100 μM) was used to the NMM-pretreated HeLa cells. In contrast, significant red fluorescence emission was observed when GSH (100 μM) was added suggesting that the probe can selectively detect GSH in the presence of Cys or Hcy. The probe was successfully applied for monitoring the GSH levels in mouse tissues such as liver, kidney, lung, and spleen.

In 2019, Xie Han et al. [89] reported an aggregation-induced emission (AIE) probe (TPE-Np, **Probe 16**, Figure 15) for selective detection of GSH. The probe is designed by modifying the widely used AIE fluorophore tetraphenylethene with a sulfonyl-based naphthalimide. Cleavage of sulfonamide by GSH induces a remarkable increase of fluorescence intensity at 496 nm which shows a good linear relationship with GSH concentrations in the range of 0–50 μM with a detection limit of 1.9 μM. The probe shows a high selectivity toward GSH than Cys and Hcy in the presence of cetyltrimethylammonium bromide. In addition, poly(ethyleneglycol)–polyethylenimine (PEG-PEI) nanogel was used as a carrier to cross-link TPE-Np to improve its solubility and biocompatibility. The probe was successfully applied for imaging and monitoring intracellular GSH level in MCF-7 cells.

##### Detection of NPSH via Cleavage of Sulfonate Ester

Dicyanomethylene-4H-pyran, one of the fluorophores with emission in red or NIR region, is featured with its better stability than other NIR fluorophores such as cyanine and squaraine. In 2014, Meng Li et al. [61] reported a colorimetric and NIR fluorescence turn-on thiol probe (**Probe 17**, Figure 16) based on dicyanomethylene-4H-pyran as the fluorophore and 2,4-dinitrobenzenesulfonyl (DNBS) as the thiol recognition group and also as the fluorescence quencher. A benzene unit was introduced to the dicyanopyran to extend its conjugated system to make its emission wavelength into the desired NIR region [90]. The fluorescence of the compound switched from off to on when the DNBS was cleaved by GSH (Ex. 560 nm, Em. 690 nm), and the fluorescence intensity increases linearly as the concentration of GSH increases (1 to 10 μM) with a detection limit of 1.8 × 10^−8^ M. A similar response was observed for other sulfhydryl-containing compounds such as Cys, Hcy, and dithiothreitol. The probe was applied for imaging GSH in HeLa cells.

In 2016, Ling Huang et al. [91] reported a soluble, distyryl boron dipyrromethene (BODIPY)-based nanomicelles probe (**Probe 18**, Figure 17) with NIR properties. DNBS was attached to distyryl boron dipyrromethene (BODIPY) to serve as a fluorescence quencher through a PET mechanism and a thiols responsive unit. The probe will be lighted up by a thiol (Cys, GSH) with an NIR emission at 660 nm (Ex. 600 nm). A PLA-PEG unit is incorporated in the structure enabling micelle formation of the compound with an excellent water solubility. The probe can quantitatively detect the thiols’ level changes in HeLa cells.

In 2018, Xiang Xia et al. [67] reported a BODIPY disulfonate probe (BODIPY-diONs, **Probe 19**, Figure 18) with a two-photon fluorescent turn-on effect for discriminately detecting GSH over Cys and Hcy. In the approach, the BODIPY dye was modified by extending the conjugated π-system to achieve a near-infrared emission. The DNBS moiety was utilized as the fluorescence quencher and also the thiol response group. A SNAr process will be initialized when the probe is exposed to GSH, resulting in the cleavage of DNBS and a significant increase of fluorescence (Em. 675 nm). Because of the double DNBS-functionalization, a different reaction rate between the probe and Cys/Hcy vs. GSH was observed with GSH being the fastest one. Later, the probe BODIPY-diONs was demonstrated to be capable of detecting GSH in live cells. In comparison with other reported fluorescent probes, BODIPY-diONs has the advantages of longer emission wavelength and low detection limit of 0.17 μM.

#### 2.1.4. Detection of NPSH via Cleavage of a Disulfide Bond

Disulfide bond (-S-S-) is another widely used thiol recognition unit for the designing of a thiol detection probe due to its minimal perturbation to the intracellular redox homeostasis. Many thiol probes were designed based on the reaction of cleavage of a disulfide bond through a thiol-disulfide exchange reaction which has been demonstrated to be thiol specific. For example, a naphthalimide derivative probe (**Probe 20**, Figure 19) made by connecting two naphthalimide units through a disulfide bond for selective and ratiometric detection of thiols were reported by Baocun Zhu et al. in 2010 [92]. The probe was switched from colorless to green color after reacting with GSH through a cleavage of a disulfide bond followed by a cyclization to release 4-aminonaphthalimide as a strong green fluorescent compound with a remarkable shift of emissions from 485 to 533 nm. The fluorescence intensity ratio of (F_533_/F_485_) (Ex. 400 nm) increased linearly with the GSH concentration in a range of 0.5 to 10 mM. The detection limit of the probe for GSH was 28 μM, enabling the ratiometric detection of GSH. The probe can successfully detect the intracellular GSH changes in live HeLa cells.

In 2016, Mingzhou Ye et al. [93] reported a probe (**Probe 21**, Figure 20) named as Cy-S-CPT (also a prodrug) in which a NIR cyanine dye and comptothecin (CPT) are connected through a disulfide bond. In the approach, the cleavage of the disulfide bond in Cy-S-CPT by thiols will induce a dramatic NIR fluorescence shift from 825 to 650 nm. In the meantime, the cleavage activates the anti-cancer drug CPT enabling a real time tracking of the prodrug distribution at 825 nm and the activated drug at 650 nm. The whole process involved the cleavage of disulfide and a cyclization to release both the active CPT and a highly fluorescent compound CyA-K. More impressively, a PEG-PLA nanoparticle loaded with the probe was able to inhibit the tumor growth rate as high as 94.0% which is much higher than the clinical approved CPT-11 (55.8%).

In 2018, Chao Yin et al. [94] developed a ratiometric photoacoustic imaging (PAI) fluorescence probe (**Probe 22**, Figure 21) for imaging GSH in vivo with a high resolution and deep penetration. The probe, IR806-pyridine dithioethylamine (PDA), employs a disulfide bond as the thiol recognition group and NIR cyanine as the fluorescence report unit. Upon reaction with GSH, the disulfide is cleaved followed by a subsequent intramolecular reaction to form a sulfydryl group leading to a ratiometrically signal changes of NIR-absorption peak from 658 to 820 nm. The detection limit for GSH is as low as 0.86 × 10^−6^ M. The probe was successfully applied for in vivo ratiometric PAI of upregulated GSH in the tumor of mice.

#### 2.1.5. Detection of NPSH via Other Strategies

In addition to the thiol detection probes presented above, numerous other thiol detection probes have been developed based on other mechanisms or strategies [30,62,95]. Although not covered in this review, there are also a number of thiol detection probes which were developed based on nanoparticles and nanocomposites, and metal ion displacement and coordination [46]. These different strategies also emerged as promising strategies for detection of thiols.

### 2.2. Selective Detection of Cys and Hcy

Cysteine (Cys) is involved in many biological processes, such as protein synthesis, detoxification, metabolism, and post-translational modification [61]. Abnormal Cys level has been linked with many diseases such as slow growth, Alzheimer’s disease and cardiovascular diseases et al. Hcy is associated with activation of multiple signal pathways and is found to be associated with cardiovascular diseases [61]. Accurate and effective detection of Cys and Hcy are essential for further exploring the roles these thiols in the biological system, A tool that can measure Cys/Hcy is in need. Although many fluorescent probes have been developed for thiols, special probes capable of discriminating Cys, Hcy, and GSH are still limited. The high GSH concentration (1–10 mM) makes it even more challenge to detect Cys and Hcy since the concentrations of Cys (30−200 μM) and Hcy (5–15 μM) are in μM.

Nevertheless, many Cys selective detection probes have been reported based on various mechanisms. These mechanisms include native chemical ligation (NCL) [96,97], aromatic substitution–rearrangement [98], and cyclization with aldehydes or acrylates [43,99,100,101]. The fluorophores employed in these probes include curcumin [38], coumarin [102,103], BODIPY [98,104,105], naphthalimide [106], rhodamine [107], fluorescein [108], and pyrene [43]. In the following, some of the mechanisms used to distinguish Cys/Hcy are presented. Most of the probes take two steps to achieve the selective detection of Cys vs. Hcy. Step one usually is the same for all the thiols (GSH, Cys, or Hcy) that is a reaction between the highly nucleophilic thiolate and a probe to form a thiol adduct. Step two commonly involved an intramolecular rearrangement reaction or cyclization reaction which occurs for Cys/Hcy adducts or Cys adduct only. Many selective probes for Cys or Hcy were developed based on the extended version of these two steps although some other probes achieve the discrimination of Cys and Hcy by the kinetic difference of the reaction between the different thiols and probes.

#### 2.2.1. Selective Detection of Cys and Hcy through Cyclization of Cys/Hcy with Acrylates or Aldehydes

Cyclization of Cys/Hcy with aldehydes or acrylates is one of the most extensively used mechanism for selectively detection of Cys/Hcy over GSH. As mentioned above, a two-step reaction takes place for the selective detection of Cys/Hcy. The method starts with a Michael addition reaction from the thiolate of a thiol (GSH, Cys, or Hcy) to form a thioether. The thioether derived from Cys or Hcy will go through a cyclization to produce a seven members-ring cyclic amide (Cys) or eight-member ring cyclic amide (Hcy) at a different reaction rate while the thioether derived from GSH is not able to undergo the cyclization reaction. Many probes have been designed based on this mechanism to selectively detect Cys/Hcy over GSH (Figure 22). For example, Xiaofeng Yang et al. [99] reported in 2011 a benzothiazole derivative which utilized an α, β-unsaturated carbonyl recognition as the thiol responsive unit to selectively detect Cys.

Zhiqian Guo et al. [102] developed a ratiometric near-infrared fluorescent probe (CyAC, **Probe 23**) for selective detection of Cys over Hcy and GSH. As depicted in Figure 22, CyAC is composed of a NIR cyanine fluorophore and an acrylate group to serve as the thiol response unit. A significant optical property change (from emission 780 nm to emission 570 nm) was observed upon reaction with Cys, while GSH and Hcy were not able to induce the change confirming the selectivity of the probe for Cys. Further, a linear relationship was found between the fluorescence intensity ratio I_560 nm_/I_740 nm_ and Cys concentration in the range of 0–25 μM. The different kinetic of the cyclization reaction was the reason for inducing the difference between Cys and Hcy, since the seven-membered ring (Cys) should be formed easier and faster than eight-membered ring (Hcy). Under the same condition, GSH was not able to induce the cyclization reaction at all.

Colorimetric probes are another widely used reagents which enable the recognition of analytes by naked-eye and do not need an instrument [47,109]. Lanfang Pang et al. [38] published their work for development of a new curcumin-based fluorescent and colorimetric probe (CAC, **Probe 24**) for specific detection of Cys. Curcumin is used because of its excellent optical properties and great biocompatibility [110,111]. As depicted in Figure 23, CAC was made by incorporating two acrylate groups with curcumin to serve as thiol recognition sites. The CAC is easier to synthesize. The detection starts from a Michael addition reaction between the thiolate of Cys and the α, β-unsaturated ketone of CAC followed by a cyclization to produce a seven membered-ring cyclic amide and a free curcumin molecule [44,112,113]. Unlike other fluorescence probes, CAC itself carries a strong fluorescence (ΦF = 0.20) which will be quenched after being activated by Cys (ΦF = 0.02), whereas the fluorescence quenching effect was not observed with other thiols like Hcy and GSH. The probe itself is colorless but can be turned into yellow color by Cys which could be easily visualized by naked eyes. In vitro cell experiments in PC12 cells showed that the probe was able to detect the change in Cys levels in live cell. The probe was further successfully applied for detecting Cys status on zebrafish.

Yao Liu et al. reported [44] a ratiometric fluorescent probe (**Probe 25**) for specific detection of Cys based on a visible-light excitable excited-state intramolecular proton transfer (ESIPT) dye (4′-dimethylami- no-3-hydroxyflavone) as the fluorophore. As illustrated in Figure 24, an acrylate group was anchored to the fluorophore to block the ESIPT and consequently quench the fluorescence. Recognition and cleavage of acrylate moiety by Cys will restore the ESIPT process and induce a remarkable fluorescence enhancement and emission wavelength shift (471 to 550 nm). The “off to on” fluorescence switch starts from a Michael addition reaction between the thiolate of Cys and the α, β-unsaturated ketone of the acrylate group followed by an intramolecular cyclization to produce a seven membered-ring amide. A kinetic difference of the intramolecular cyclization resulting in a selective detection of Cys over Hcy and GSH. A steric effect could also possibly elevate the kinetic difference of the reaction from Cys over Hcy. Upon completion of the reaction, a nearly 40-fold increase of ratiometric value of the emission intensities (I_550_/_471_) was induced. Additionally, the probe was able to detect Cys level changes in HeLa cells. Additional advantages of the probe include a large Stokes shift (>130 nm), fast response (within minutes) and high sensitivity (~0.2 μM) for Cys.

Similar to the structure of acrylate, a bromoacetyl group was also used to develop probes for detection of Cys/Hcy employing the same sensing mechanism. In 2013, fluorescein chemodosimeter was developed by Keum-Hee Hong et al. [114] as a fluorescent probe (**Probe 26**, Figure 25) for detection of Cys over Hcy and GSH. The reagent was made by introducing a bromoacetyl group to fluorescein. The selectivity for Cys over Hcy and GSH in aqueous solution is due to the thermodynamically stable and kinetically rapid formation of six-member lactam ring in aqueous solution with Cys [115]. The reagent was capable of detecting Cys changes in HeLa cells.

Hye Yeon Lee et al. [43] reported two new pyrene-based fluorescent probes (**Probes 27**, **28**, Figure 26) for selective detection of Hcy over Cys and other thiols. The two probes were converted rapidly (10 min) to a high fluorescence (350 nm and 450 nm) form when mixed with Hcy, while no fluorescence response was observed when mixed with Cys and GSH even over a long reaction time. The detection limit of Hcy was determined to be 1.44 × 10^−7^ M with a linear range of 600–1000 nM in HEPES containing 10% DMSO (0.01 M and pH 7.4). The authors provided an explanation for the fluorescence response difference for Hcy over Cys and GSH. These probes have been applied for the detection of Hcy changes in mammalian and showed no response to the change of GSH.

#### 2.2.2. Selective Detection of Cys and Hcy through Cys-Induced SNAr Substitution−Rearrangement Reaction

The Cys-induced SNAr substitution−rearrangement reaction used for selective discrimination of thiols has been widely explored. Many probes have been developed based on various fluorophores that possess the ability to undergo a *S* to *N* displacement rearrangement reaction presented early. The fluorophores employed include BODIPY, NBD, naphthalimide, and cyanine (Figure 27). The requirements for fluorophores to be applied for the purpose are that they should be able to form a thioether (GSH, or Cys/Hcy) first. Then the amino group of Cys/Hcy conjugated thioether will rearrange by displacing the sulfur of the thioether with the amino group to form amino-substituted product. As described early, the rearrangement does not occur with GSH thioether. An example of the mechanism is illustrated in Figure 28. The remarkable optical property differences between sulfur and amino substituted BODIPY enable the discrimination of GSH over Cys/Hcy.

##### BODIPY

BODIPY fluorescent dyes featured with a high molar absorption coefficient, quantum yield, and photo stability which enable them to be widely applied in many fields [116].

In 2012, Liya Niu et al. [29] developed monochlorinated boron dipyrromethene (BODIPY) derivatives to selectively detect GSH over Cys/Hcy. However, the emission band of amino-substituted BODIPY (Cys and Hcy) is close to the emission band of the original monochlorinated boron dipyrromethene (BODIPY) probe, Thus, the probe failed to detect Cys and Hcy. In 2013, a structure modification [117] by adding a nitrothiophenyl or nitrophenyl group to the same BODIPY fluorophore was employed to address the problem (**Probe 29**). As depicted in Figure 29, the probe itself displays no fluorescence (PET) but can be turned on to fluorescent products upon reaction with thiols followed by the occurrence of the rapid intramolecular displacement of sulfur with the amino group of Cys and Hcy. The reaction of Cys is much faster than Hcy while no such intramolecular displacement occurred for GSH achieving the ability of discrimination of Cys from Hcy and GSH. The probe itself shows barely any fluorescence but can be turned on to strong fluorescence by thiols with the product derived from Cys exhibiting emission wavelength at 564 nm while the products derived from GSH and Hcy emitting at 588 nm. The fluorescence intensity at 565 nm followed a nicely linear relationship with the Cys concentration (0–100 μM) with a detection limit of 2.12 × 10^−7^ M. The probe was successfully shown to be capable of detecting Cys status change in live cells.

##### 7-Nitrobenzofurazan (NBD)

7-Nitrobenzofurazan (NBD) is another widely used and extensively studied fluorophore that is able to go through the Cys-induced SNAr substitution−rearrangement reaction for selective detection of thiols (Figure 28).

In 2014, Liya Niu et al. [118] explored NBD as another fluorophore that can go through the intramolecular displacement mechanism to discriminatively detect different thiols. As shown in Figure 30, Cys/Hcy and GSH form different release fluorophores NBD, *N*-substitute products (Cys/Hcy) or *S*-substitute product (GSH). The *N*-substitute product (Cys/Hcy) exhibit significantly higher fluorescence than the *S*-substitute product (GSH). A simple probe NBD-Cl (**Probe 30**) was developed and successfully applied for detecting Cys in live cells. Pinaki Talukdar et al. [119] found the same phenomenon which further push the development of Cys/Hcy selective probes based on NBD.

In 2019, Lihui Zhai et al. [120] developed a dual emission turn on fluorescent probe (CA-NBD, **Probe 31**) for discriminative detection of Cys/Hcy and GSH simultaneously based on NBD. The probe was designed by combing another fluorophore coumarin together with 7-nitrobenzofurazan (NBD). Two different NBD derivatives formed, one is *N*-substitute products (Cys/Hcy) and the other is S-substitute product (GSH). At the excitation wavelength of 460 nm, GSH-NBD (*S*-substitute) derivative was non-emissive. In contrast, the Cys/Hcy-NBD derivatives exhibited strong yellow fluorescence enabling the discrimination of Cys/Hcy over GSH. The fluorescence intensity was linearly increased when the thiol concentration with a detection limit of 2.00 × 10^−8^ M for Cys and 1.02 × 10^−8^ M for Hcy, respectively. The probe can successfully detect the Cys status change in live cells.

A dual channel responsive NIR fluorescent probe (**Probe 32**, Figure 31) for selective detection of Cys in live cells was reported by Zhuo Ye et al. [121] in 2017. The probe named as BDY-NBD is composed of two fluorophores, a NIR BODIPY fluorophore, BDY-OH, and a NBD fluorophore. In the structure, NBD is employed as the thiol responsive unit and the fluorescence of BDY-OH is quenched by NBD owing to PET. Once the probe reacts with a thiol, the nucleophilic thiolate will cleave the NBD from BDY-OH and attach to NBD to form a NBD thiol adduct. The NIR fluorescence of BDY-OH (λex = 650 nm, λem = 735 nm) will be recovered because of the termination of PET. Different from GSH, the thiol adduct (Cys-NBD) produced in the reaction between Cys and probe will subsequently undergo an S to N acyl shift reaction to form a much stronger fluorescent product with a much different emission wavelength (Ex. 466 nm, Em. 540 nm) than BDY-OH while the shift reaction does not occur for GSH. Therefore, a dual-emission mode can be utilized for the detection of Cys. The probe has a much low detection limitation (22 nM, 540 nm emission) that is lower than most of the fluorescent probes developed for the detection of Cys previously [100,122,123]. In vitro cell experiments with HeLa cells demonstrated that the probe can be applied for multicolor imaging of intracellular Cys, and a significant differences of fluorescence imaging were observed for GSH and Cys in live cells when their levels are manipulated.

Many other probes have been designed and applied based on different thiol recognition groups or other modifications. Shuangshuang Ding et al. [124] developed the probe (**Probe 33**, Figure 32) by combing a benzothiazole molecule and 7-nitrobenzofurazan (NBD) molecule through an ether bond. Longwei He et al. [125] designed and synthesized a probe (**Probe 34**, Figure 32) by attaching NBD to another fluorophore hydroxyphenyl benzothiazole merocyanine (HBTMC) through an ether bond. The probe was able to detect Cys/Hcy, GSH, and H_2_S at different emission wavelengths. Peng Wang et al. [126] developed a NIR probe (**Probe 35**, Figure 32) named as DCM-NBD which was made by combining a NIR fluorophore icyanomethylene-4*H*-pyran (DCM) derivatives and a NBD fluorophore via ether linkage. The probe was successfully being applied not only in vitro but also in animals.

Similar with the structure of NBD, a nitrobenzothiadiazole probe (**Probe 36**, Figure 32) was also developed by Dayoung Lee et al. [127] to achieve the goal of selective detection of Cys and Hcy over GSH based on the same mechanism. The probe was found to increase fluorescence at pH 7.4 in the presence of Cys or Hcy, while at weakly acidic conditions (pH 6.0), only Cys can induce the fluorescence enhancement.

##### Other Fluorophores

In addition to NBD, BODIPY, probes based on naphthalimide, and cyanine [118] have also been extensively developed, such as 4-nitro-1,8-naphthalic anhydride (NNA, **Probe 37**, Figure 33) which was reported by Limin Ma et al. [128] in 2012 based on nitro-naphthalimides as the fluorophore and aromatic nitro part as the thiol recognition group. The probe can selectively detect Cys with a detection limit of 0.3 μM by following the same mechanism (Figure 28) exhibited for NBD and BODIPY.

#### 2.2.3. Selective Detection of Cys and Hcy via Other Mechanisms

Many other probes have been developed for discrimination of Cys/Hcy over GSH based on different sensing mechanism. For instance, Hyo Sung Jung et al. [129] developed a probe (**Probe 38**, Figure 34) in 2011 that can selectively detect intracellular Cys by the Michael addition reaction in combination with a steric hinderance factor. A bulky substituent was added to achieve the preferential response for Cys relative to GSH since the molecule of GSH or Hcy is larger than Cys. In addition, the lower pKa of Cys than Hcy and GSH is another factor leading to a higher nucleophilicity of Cys. The fluorescence intensity was found to be linearly increased with an increase in Cys concentration in a range of 0–0.9 mM. The detection limit was determined to be 10^−7^ M for Cys. Later, the probe was successfully applied for monitoring intracellular Cys in HepG2 cells. The fluorescence derived from the probe in live cells decreased after an increase in NEM concentration. Xin Zhou et al. [130] developed a Cys selective detection probe (**Probe 39**, Figure 34) based on a Michael addition reaction assisted by an electrostatic attraction.

While most of the probes being developed are capable of sensing only one of the thiols at a time, a probe which can selectively detect two or three thiols simultaneously with different emissions is highly desirable. In order to address the need, Jing Liu et al. [102] developed a chlorinated coumarin-hemicyanine probe (**Probe 40**, Figure 34) which can simultaneously detects Cys and GSH separately from two different emission and excitation channels. Similar with the two steps strategy for discrimination of thiols, this new coumarin-hemicyanine dye utilize the same strategy starting from a SNAr nucleophilic substitution first to produce a thio-coumarin-hemicyanine, followed by a different intramolecular rearrangement reaction (GSH vs. Cys/Hcy) or different kinetic of the same intramolecular reaction (Cys vs. Hcy) to selectively detect GSH or Cys. The probe is non-emissive and can be turned on by Cys and GSH with emission maximum at 420 and 512 nm, respectively, indicting the high selectivity of the probe for Cys or GSH. For both Cys and GSH, a linear relationship was found between the fluorescence intensity at their emission wavelength and their concentrations (GSH, 0 to 0.9 equiv; Cys, 0.4 to 0.8 equiv). The probe was demonstrated to be able to simultaneously monitor Cys and GSH in COS-7 cells in multicolor imaging.

## 3. Detection of Thiols in Subcellular Organelles

Thiols are known to distribute unevenly in subcellular organelles [8] such as mitochondria, lysosomes, endoplasmic reticulum (ER), golgi apparatus, nucleus, and cell surface. Their levels in different organelles could be an indicator or biomarker to reflect the status of the organelles. Thus, tools or methods that can map thiol distribution and monitoring their status changes not only for the whole cell but also for subcellular organelles will provide valuable information to understand the relationship between thiols and cells’ functions and dysfunctions, and reveal their correlation as well as find treatment for various thiol-associated diseases [9]. While extensive work has been made and plenty of probes have been developed for the detection of thiols for the whole cell, efforts have been made to develop tools for detecting thiols in subcellular organelles. A subcellular organelle targeting structure is usually needed for a subcellular thiol imaging probe to direct the probe into the targeted organelle. A few subcellular organelle targeting thiol probes have been developed based on the difference identified for subcellular organelles. Most of these probes are for mitochondria with a few for lysosome and ER. In the next part, the diverse targeting approaches and sensing mechanisms of subcellular organelle-targeted thiols probes will be discussed and summarized.

### 3.1. Mitochondrial Total Thiols and GSH

As one of the most important subcellular organelles in cells, mitochondria are double-membrane constructed organelles that serve as the energy generator and also involved in many other biological process including calcium circulation, protein synthesis, apoptosis pathways, etc. A large amount of reactive oxygen species (ROS) are generated in mitochondria during the energy generation process [131,132,133]. With thiols (especially GSH) being the most important antioxidant to terminate ROS in cells, it is of importance to have a tool that can monitor mitochondrial thiols in live cells. Inspired by the knowledge that mitochondrial membrane spans carry a negative charge [134], a lipophilic cationic structure are widely used for achieving the mitochondria targeting. These lipophilic cations include triphenylphosphonium (TPP), positive charged cyanine, and rhodamine (Figure 35). In addition to lipophilic cations, functional groups, such as peptides [135] have also been developed for mitochondria targeting [136]. In this section, probes developed for detection of mitochondria thiols will be illustrated based on their sensing mechanisms.

#### 3.1.1. Detection of Mitochondrial Thiols via a Michael Addition Reaction

In 2020, Yutao Yang et al. [137] developed a NIR fluorescent probe (NIR-HMPC, **Probe 41**, Figure 36) based on a Michael addition reaction followed by self-immolative reaction to selectively detect mitochondrial thiols. This NIR probe made by conjugating hemicyanine dye with a benzopylium was connected with a 7-hydroxymethyl-2,3- dihydro-1H-cyclopent-a[b]chromene-1-one with a carbonyl ester. The positive charged indoles iodized salt in the molecule attributes to the mitochondria selectivity. Upon exposure to thiols, nucleophilic addition of the sulfhydryl group will lead to the chromene ring-open to result in a released phenol anion. Subsequently, phenol anions will trigger a self-immolative reaction to generate carbon dioxide and release a strong fluorescent NIR product (λ_em_ 731 nm). The fluorescence intensity (731 nm) was found to be linearly increased as the concentration of GSH, Cys, and Hcy increased with the detection limit of 0.59 μM, 0.39 μM, and 0.54 μM, respectively. Colocalization studies with DAPI and Mito-Tracker Green in MCF-7 cells, HepG2 cells, and HeLa cells clearly confirmed that NIR-HMPC can effectively imaging mitochondrial thiols. When PMA (Phorbol 12-myristate 13-acetate, a cellular oxidative stress inducer) and BSO (a sulfoximine that can reduce GSH levels) were used to pretreat cells to induce an intracellular thiol decrease, a dramatic fluorescence decrease was observed revealing that the probe was able to visualize and reflect mitochondrial thiol change in live cells. More impressively, the probe can not only reflect thiol change in vitro but also reflect the change in vivo in a mouse model. The probe has also been successfully applied on sensing thiol change in a mouse mode of cerebral ischemia, warranting the application of the probe for monitoring the physiological and pathological processes at the cellular and animal levels.

In 2017, Jianwei Chen et al. [138] developed a new mitochondrion-targeted probe (MitoRT, **Probe 42**, Figure 37) which could detected the mitochondrial GSH status reversibly. In the molecule, TPP, the positive charged mitochondria targeting group, was linked to a fluorophore via an optimized 4-carbon linker. The reagent is capable of reacting with GSH rapidly in both forward and reverse directions that enable the real time monitoring of mitochondrial GSH dynamically in live cells. In addition, MitoRT showed ratiometric fluorescence responses with a wide dynamic range when reacting with GSH. The probe has been demonstrated to be capable of monitoring mitochondrial GSH in live HeLa cells reversibly through fluorescence microscope. The probe can also be applied to monitor mitochondrial GSH levels in a high throughput manner (flow cytometry).

In 2017, Keitaro Umezawa et al. [139] reported their work on developing reversible fluorescent probe (**Probe 43**) for live cell imaging and ratiometric quantification of fast GSH dynamics in mitochondria based on a cationic rhodamine fluorophore as an electrophile for Michael addition to reversibly react with thiols. The calculated rate constant *k* = 560 M^−1^ s^−1^ (t_1/2_ = 620 ms at GSH = 1 mM) which is a 3900-fold increase than TQ green [73]. Similar with GSH, other thiol-containing molecules such as Cys, Hcy, and H_2_S can also induce the fluorescence change if their concentration is in millimolar range. However, since their physiological concentration is much lower than GSH, their interference with GSH should be a minimum. As shown in Figure 38, a TMR (typical rhodamine structure) was introduced to the probe as a donor of the FRET. Based on that, two probes, QG0.6 and QG3.0 were designed and synthesized. Both probes exhibit ratiometric absorption/fluorescence changes in response to various concentrations of GSH. Co-staining experiments with MitoTracker Deep Red FM indicated that the compound could accumulate inside mitochondria. QG3.0 can quantitatively monitor GSH status change in real time when cells are consecutively treated with H_2_O_2_ or GSH ester. The probe is also capable of monitoring the real time GSH level change after the cells being glucose deprived.

#### 3.1.2. Detection of Mitochondrial NPSH via a SNAr Reaction Using Halogen, Ether, or Thioether as a Leaving Group

##### Detection of Mitochondrial NPSH via a SNAr Reaction Using Halogen as a Leaving Group

In 2017, Xueliang Liu et al. [140] reported a mitochondria-targeting fluorescent probe (BODIPY-PPh_3_, **Probe 44**) for selective and ratiometric detection of GSH over Cys and Hcy in line with the group’s previous work [29]. As depicted in Figure 39, chlorinated BODIPY was used as the fluorophore and the chlorinated site serves as thiol (GSH, Cys, or Hcy) recognition site through a SNAr reaction to afford the thioether. As discussed early, the thioether formed with Cys and Hcy underwent an intramolecular rearrangement to yield the amino derivatives resulting in a remarkable change of photophysical properties, while the thioether formed with GSH could not undergo such an intramolecular rearrangement. The difference in the intramolecular rearrangement reaction for GSH vs. Cys and Hcy attribute to the selective detection of GSH over Cys and Hcy. To achieve mitochondria targeting, a TPP structure was attached to the probe. The probe itself shows an intrinsic emission at 557 nm which decreases dramatically upon reaction with GSH and a new emission at 588 nm (Ex. 550 nm) increased significantly. Their ratio (*I*_588_/*I*_557_) was found to increase linearly with GSH concentration (0–80 μM) with a detection limit of 1.1 μM. The linearity enables the quantitative detection of GSH. The fluorescence behavior of the probe reacting with Cys and Hcy was totally different with a much lower emission at 588 nm and much smaller (*I*_588_/*I*_557_) ratio change. The colocalization study with rhodamine 123 found that BODIPY-PPh_3_ could accurately locate in the mitochondria. Pretreatment of the cells with NEM to deplete GSH resulted in a lower red/green fluorescence ratio (~2). In contrast, pretreatment of cells with GSH before incubation with BODIPY-PPh_3_ led to a higher red/green channels fluorescence ratio (~5). The results reveals that the BODIPY-PPh_3_ can effectively and ratiometrically monitor the status of mitochondrial thiols.

In 2018, Sujie Qi et al. [141] presented a water-soluble ratiometric fluorescent probe (**Probe 45**) for selective detection of GSH in mitochondria of live cells. The probe was built based on colorimetric hemicyanine dye and was capable of discriminating GSH over Cys/Hcy in a quantitative manner in a range of 1.0–15.0 mM with a low detection limit in aqueous solution. In detail, the hemicyanine dye was modified by an aldehyde group and a chloro-substitution which equip the probe the ability to selectively react with GSH since GSH reacts with the probe via a SNAr and an intramolecular aldimine condensation to form a ring (Figure 40). The reaction was much faster than the reaction with Cys and Hcy since an intramolecular aldimine condensation to form a ring could not occur with Cys and Hcy [65,142]. The quaternary ammonium cation serves as the mitochondria targeting moiety and a hydrophilic sulfonate was added to enhance the water-solubility of the probe. Upon reaction with GSH, significant optical property (absorption and fluorescence) changes were induced which significantly dropped fluorescence intensity at 607 nm along with appearance of a new fluorescence peak at 648 nm. The ratio of emission intensities (I_648_/I_607_) displayed a linear relationship with GSH concentration (1.0 to 15.0 mM) with a detection limit of 24.16 μM, enabling the quantitative detection of GSH. The probe displays a great mitochondria targeting ability and can well reflect the status changes of mitochondrial GSH in a ratiometrical manner.

##### Detection of Mitochondrial NPSH via a SNAr Reaction Using Thioether as a Leaving Group

In 2018, we developed a mitochondrial-targeting rhodamine based probe [22] (TBROS, **Probe 46**) for selective imaging and quantification of mitochondrial thiols in live cells. The probe was designed based on a thiol specific thiol–sulfide exchange reaction reported by us in 2012 [42]. Similar to the GUALY’s reagent [42], two benzofurazan moieties were connected together by a thioether to maintain a symmetric structure (Figure 41). Two cationic rhodamine B units were introduced symmetrically to the benzofurazan structure to provide the mitochondria selectivity and also serve as a strong fluorophore to enhance the fluorescence intensity of benzofurazan. The probe itself is not fluorescent due to the self-quenching of the symmetric structure but turns into strong fluorescent products (Ex. 550 nm, Em. 580 nm) upon reaction with thiols through the thiol specific thiol–sulfide exchange reaction. TBROS has been confirmed nonreactive toward PSH and successfully used to image and quantify NPSH in mitochondria in live cells.

In 2019, Zhiqiang Xu et al. [143] developed a NIR probe (Cy-S-Np, **Probe 47**, Figure 42) based on thioether as the thiol response unit to visualize mitochondrial and lysosomal GSH in live cells and also in a mouse model. The probe was constructed by connecting a cationic cyanine IR-780 dye (mitochondria targeting) and a morpholine-coating naphthalimide (lysosome targeting) through a thioether (thiol response moiety). Cy-S-Np can effectively distinguish GSH over Cys and Hcy in mitochondria and lysosome, warranting its promising use for studying the interaction of cellular function between mitochondria and lysosome. After reaction with GSH, a remarkable fluorescence intensity enhancement at 812 nm (Ex. 710 nm) was observed. Impressively, the probe showed a detection limit of 11 nM. Cellular experiments showed that the probe was able to detect GSH changes both in mitochondria and in lysosome of live HeLa cells. Further, visualization of subcellular GSH level at organism level in mice was achieved as well.

##### Detection of Mitochondrial NPSH via a SNAr Reaction Using Ether as a Leaving Group

Yuan Gu et al. [35] developed a ratiometric reagent which can be used in two-photon fluorescence microscopy (TPFM). The two photon excitation fluorescent imaging has the advantage of less interference from autofluorescence background, deeper penetration to tissues, and low phototoxicity [144,145,146,147]. The regent (**Probe 48**, Figure 43) named as TPE-PBP showed a high sensitivity and selectivity toward thiols, including GSH, Cys, Hcy, etc., with a high selectivity for mitochondria. Cell experiments shows that TPE-PBP was able to measure the mitochondrial thiol status change in a ratiometric manner. The two-photon-absorption cross section enables the regent to apply in live cells, in living skeletal muscle tissues, and in two day old fish larva. The success of this reagent provides a new strategy for the construction of ratiometric two-photon active reagens on the application of in vivo biosensing and bioimaging application [35]. It needs to be noted that the application of traditional fluorophore probes was hampered by the aggregation-caused quenching (ACQ) effect that is that the probes’ fluorescence will be quenched or weakened if the probes are concentrated or aggregated [148,149,150]. The ACQ effect makes the quantification of analytes in live cells more challenging. Tang and co-workers observed an unusual phenomenon on a class of molecules with propeller shape (e.g., tetraphenylethylene (TPE), siloles) which can turn on their fluorescence in the aggregate or solid states while keep non-emissive in diluted solutions. This phenomenon was termed as “aggregation-induced emission (AIE)” and the mechanism can be explained by a restriction of intramolecular motions (RIMs), including restriction of intramolecular vibrations (RIVs) and restriction of intramolecular rotations (RIR) [151,152,153,154,155,156,157,158,159]. Herein, AIE probes can overcome the drawbacks of the traditional fluorophore probes.

In 2016, Jian Zhang et al. [160] developed a probe (**Probe 49**, Figure 44) for detecting mitochondrial GSH with BODIPY serving as the fluorophore. In this approach, a self-immolative dinitrophenoxy benzyl pyridinium was connected with BODIPY to serve as the targeting ligand and also the GSH recognition group. Once being exposed to GSH, the dinitrophenyl moiety will be cleaved through a SNAr reaction, followed by a self-immolation reaction to release fluorophore BODIPY with strong fluorescence (599 nm). A linear relationship was observed between the fluorescence intensity and GSH concentration in a range of 1 to 15 μM with a detection limit of 109 nM. The selectivity of GSH over Cys and Hcy was possible due to an electrostatic interaction between the cationic pyridinium moiety and GSH: once the electrostatic interaction formed, GSH was believed to react much faster than Cys and Hcy. The probe has been successfully used to monitor the status of mitochondrial GSH in live HeLa cells.

In 2019, Yue Xu et al. [1] reported a visible and near-infrared, dual emission fluorescent probe (Cy-DC, **Probe 50**) for monitoring mitochondrial thiol in vitro. The probe also exhibits impressive ability to detect solid tumor by both naked eye and near-infrared fluorescence. As depicted in Figure 45, two fluorophores, dicyanomethylene-4H-pyran and cyanine, with distinct wavelength emission bands are linked together with a thiol responsive unit ether linker aryl ether group. The positive charged cyanine is the driving force for mitochondria targeting. The fluorescence activation by thiols was achieved with a nucleophilic aromatic substitution-rearrangement reaction between the thiolate of a thiol with the aryl ether to form a thioether which the thiol molecule is attached to the cyanine with strong NIR fluorescence (Em. 810 nm). In the meantime, the other fluorophore, dicyanomethylene-4H-pyran, was released and restored its visible fluorescence (520 nm). In contrast to GSH, the thioether formed from Cys and Hcy will undergo an intramolecular rearrangement vis a 5-(Cys) or 6-(Hcy) membered transition state resulting in the replacement of *S* with *N* leading to a significant fluorescence elimination (810 nm). This provides the probe the ability for discrimination of GSH over Cys and Hcy. Briefly, GSH activated a remarkable increase of fluorescent intensity of Cy-DC both at 520 nm and 810 nm while Cys and Hcy only attribute to the fluorescent intensity of Cy-DC at 520 nm. The detection limits for GSH in two different channels were determined to be 24 nM (visible) and 32 nM (NIR). Additionally, an increase in the thiol blocking agent (NEM) led to a gradually decrease of fluorescence intensity in both channels. In line with the hypothesized mechanism, pretreatment of cells with NEM totally inhibited the fluorescence while addition of GSH to the cells induced a remarkable increase in fluorescence intensity in both channels (green visible channel and NIR red channel). In contrast, addition of Cys and Hcy only led to the fluorescence increase in green visible channel while NIR red fluorescence intensity was barely changed.

In 2019, Mingming Cui et al. [161] reported a turn on fluorescent probe (**Probe 51**, Figure 46) for detection of mitochondrial GSH with twisted intramolecular charge transfer (TICT) and aggregation-enhanced emission (AEE) characteristics. Cleavage of the dinitrophenyl ether from QUPY-S by GSH followed by a self-immolation reaction results in a low water solute compound QUPY which aggregates to turn on the fluorescence at 516 nm. The detection limit was determined to 434 nM. The probe is featured with a large Stokes shift (131 nm) and capable of detecting GSH in HeLa cells.

#### 3.1.3. Detection of Mitochondrial NPSH via Cleavage of Sulfonamide or Sulfonate Ester

##### Detection of Mitochondrial NPSH via Cleavage of Sulfonamide

In 2018, Zhiqiang Xu et al. [162] developed a visible and near-infrared, dual-channel fluorescence-on robe (CyP-SNp, **Probe 52**, Figure 47) for monitoring mitochondrial GSH in a spatiotemporal and synchronous manner. In the approach, two widely used strong fluorophores, naphthalimide (a visible fluorophore) and cyanine (a NIR fluorophore), are bridged together by a thiol-reactive sulfonamide moiety. These two fluorophores were chosen since they are emitting in two distinctly different wavelength regions. The probe which displays a weak fluorescence will be turned on to two completely different fluorescence upon reaction with GSH, green fluorescence (Em. 495 nm, Ex. 370 nm) in the visible channel and red fluorescence (Em. 795 nm, Ex. 700 nm) in the near-infrared channel. The cationic cyanine dye accumulates in mitochondria. The detection mechanism involves a nucleophilic substitution reaction between the thiolate of GSH with sulfonamide to form an intermediate and subsequent cleavage of the sulfur–nitrogen bond of the intermediate will result in a stable NIR fluorescent product CyP and another reactive intermediate Np-GSH-SO_2_ which will switch to a visible-light-emitting product Np-GSH after releasing SO_2_. Cys and Hcy follow the similar mechanistic pathway in a much slower manner. The kinetic differences enable the probe to discriminatively detect GSH over Cys and Hcy. The detection limits in the two different channels are 1.53 × 10^−7^ M (visible channel) and 1.71 × 10^−7^ M (near-infrared channel) respectively. Cellular experiments showed that the probe display good mitochondria-targeting capacity.

The probe was further demonstrated to be capable of monitoring GSH status in live cells with a minimum interference from Cys and Hcy. Results showed that addition of Cys and Hcy did not change the fluorescence intensity significantly while addition of GSH induced a remarkable intense fluorescence increase. Later the probe showed to be capable of tracking GSH levels in living tissues with imaging depths of up to 120 μm in the near-infrared channel [163].

In 2020, Zhengkun Liu et al. [164] reported a two-photon probe MT-1 (**Probe 53**, Figure 48) based on a two-photon and fluorescence resonance energy transfer (FRET) strategy. In the molecule, naphthalimide was used as the two-photon receptor as well as the FRET donor while a rhodamine B group was used to serve as the FRET acceptor as well as the mitochondria-targeting ligand. DNBS was attached to naphthalimide moiety via piperazine and utilized as thiol recognition group. In addition, DNBS is a high electron withdrawing group (EWG) which could withdraw the electron to induce intramolecular charge transfer (ICT) effect resulting in the quenching of the fluorescence [165,166]. More importantly, this is the first probe developed for mitochondrial thiols based on two-photon and Förster resonance energy transfer (TP-FRET) strategy, and the probe was successfully applied for detection of mitochondrial thiols in live cells and in mouse liver tissue slices.

##### Detection of Mitochondrial NPSH via Cleavage of Sulfonate Ester

In 2018, Fangfang Wang et al. [167] presented a BODIPY-based fluorescent probe (**Probe 54**) for fast detection of intracellular mitochondrial thiols with high sensitivity and selectivity by introduction of a dual reactive group. Although BODIPY exhibits a number of advantages as a fluorophore, such as high fluorescence quantum yield, excellent photostability, and intense absorption of visible light [160,168,169,170], utilization of BODIPY for building a BODIPY-based turn-on fluorescent probes is hindered by the hardship of completely quenching the original fluorescence of BODIPY leading to a high background and low sensitivity. To overcome the challenge, two quenching groups, a DNBS moiety and a nitroolefin moiety (-CH=CH-NO_2_) were introduced to the BODIPY fluorophore. These two groups also serve as the thiol recognition groups. In the molecule, DNBS serves as the acceptor of the PET and nitroolefin moiety (-CH=CH-NO_2_) serves as a Michael acceptor for the PET as well as a receptor for the intramolecular charge transfer (ICT) process. Thus, the fluorescence of the probe was heavily quenched by these dual quenching systems. The dual reaction of GSH with the probe (Figure 49) breaks the PET and ICT process and restores the strong fluorescence of BODIPY (Em. 517 nm). A linearly increase of fluorescence was observed with an increase in GSH concentration (0–350 μM), Hcy (0–300 μM), and Cys (0–100 μM), enabling the quantitatively detection of thiols. The detection limits of the probe for Hcy, Cys, and GSH were determined to be 87 nM, 147 nM, and 129 nM, respectively. Later, the probe was demonstrated to selectively accumulated inside mitochondria. The ability of the probe to reflect the mitochondrial thiols change was confirmed with live HeLa cells, a significant decrease of fluorescence was observed when the cells were pretreated with NEM. Medium wash was not needed in cell imaging since the probe exhibited a low fluorescence background. The probe was able to image mitochondrial thiols in ca. 10 min.

In 2017, three two-photon fluorescent mitochondrial thiol probes (**Probes 55**–**57**, Figure 50) were reported by Yi Li et al. [171] based on coumarin as the fluorophore. The probes employed an imidazolium cation to target mitochondria and a strong electron-withdrawing DNBS was introduced to serve as the thiol recognition site and to quench the fluorescence of the coumarin fluorophore via an intramolecular charge transfer (ICT) reaction. A thiol molecule will react with the probes and remove the DNBS structure to turn on the fluorescence (Em. 482 nm). The probes showed dramatic optical responses to various thiols, such as GSH, Cys, and Hcy. Moreover, the probes showed response to BSA confirming they can also react with PSH. In contrast, negligible response was observed for non-thiol amino acids. The authors demonstrated that the probes were located inside mitochondria via a colocalization study with mito-tracker red while probes without the imidazolium cation distributed everywhere inside the cells. These probes were capable of monitoring thiol level change in A549 cells. The ability of detecting thiol level changes was further confirmed in the zebrafish. The penetration depth of these two-photon probes was determined to be 160 μm in the vasculature of an anaesthetized mouse using two-photon microscopy. Overall, these probes showed high mitochondrial selectivity, large two-photon absorption cross-section, and good biocompatibility. They are promising reagents for application for thiol imaging in vitro and in vivo.

In 2020, Xin Li et al. [172] developed a novel composite NIR dye (Hcyc-NO, **Probe 58**) for effective detection of thiols and oxidative stress inside mitochondria in live cells. In the approach (Figure 51), a novel NIR hemicyanine dye (Hcyc) composed by cyanine and coumarin were designed and synthesized. The probe itself exhibits weak fluorescence but can be turned on to a high fluorescence product (Ex. 723 nm, Em. 751 nm) upon reaction with thiols, such as GSH, Cys, Hcy H_2_S, and dithiothreitol (DTT). Removal of DNBS group from the probe induced the fluorescence enhancement. The probe displays an instantaneous response (<5 s) to thiols with high selectivity and sensitivity. A linear relationship was observed when the probe reacted with various concentrations of GSH (0–5.0 μM), Cys (0–2.0 μM), and Hcy (0–1.0 μM) with the detection limits of 0.11, 0.08, and 0.20 μM, respectively. A cellular colocalization study with Mito-Tracker demonstrated that the probe was localized inside mitochondria. The probe demonstrated the ability to detect a change in thiol levels in live cells. Furthermore, Hcyc-NO was confirmed to be capable of quickly detecting the content of thiols or oxidative stress in vivo in a mouse model.

#### 3.1.4. Detection of Mitochondria NPSH via Cleavage of Disulfide Bond

In 2011, Chang Su Lim et al. [173] developed a two-photon probe (SSH-Mito, **Probe 59**) for ratiometrically detecting of mitochondrial thiols. In the approach, 6-(benzo[d]thiazol-20-yl)-2-(*N*,*N*-dimethylamino)naphthalene (BTDAN) was used as the fluorophore, TPP as the mitochondrial-targeting site, and a disulfide group as the thiol recognition group. The reaction was illustrated in Figure 52. Thiol will first react with the disulfide bond followed by a cleavage of the C–N bond to afford the final product. The process will shift the fluorescence from blue color (462 nm (Φ = 0.82)) to yellow color (545 nm (Φ = 0.12)) and the ratio of the fluorescence intensity of F_yellow_ (525–575 nm) to F_blue_ (425–475 nm) elevated 42 to 77-fold in the presence of thiols (GSH, Cys, dithiothreitol (DTT), 2-mercaptoethanol (2-ME), and 2-AET). In contrast, no significant change was observed in the presence of non-thiol amino acids. In vitro cellular experiments demonstrated that SSH-Mito predominantly accumulated inside mitochondria. The probe was proved to be able to respond to thiols’ changes in live cells. Moreover, the probe was successfully used for visualizing thiol levels in living tissue depths of 90–190 μm.

In 2020, Lu Wang et al. [174] reported a ratiometric fluorescent probe (**Probe 60**, Figure 53) for detection of mitochondria thiols. The probe was built based on the FRET strategy. BODIPY and rhodamine were used in this approach to achieve the FRET-based fluorescent probes since there is a clear overlap between the emission of BODIPY and absorption of rhodamine [175,176]. Thus, BODIPY and a modified near-infrared rhodamine [26,177] were connected together with a disulfide to build an effective FRET structure with BODIPY serving as the donor and rhodamine as the acceptor. In the meantime, the disulfide is the thiol recognition group and also the switch for turning on-off the FRET. When the probe was excited at 488 nm (BODIPY’s excitation wavelength), a strong emission with a peak at 656 nm of rhodamine acceptor was observed and a weak emission peak at 512 nm (BODIPY’s emission) due to the high efficiency FRET process. Once the probe reacted with a thiol molecule (GSH, Cys, or Hcy), the disulfide was cleaved by thiols and the FRET was turned off resulting in an increase of the emission intensity at 512 nm (corresponding BODIPY fluorescence) while the fluorescence intensity at 656 nm decreased. The fluorescence ratio (F_512 nm_/F_656 nm_) increased linearly as GSH concentration increased from 10 to 100 μM with a detection limit of 0.26 μM suggesting that probe can be used as a ratiometric fluorescent probe for quantitative detection of thiols. The probe was demonstrated to be well localized inside mitochondria and able to detect the mitochondrial thiols’ change in live HeLa cells.

In 2017, Qingbin Zeng et al. [178] developed a mitochondria targeting and intracellular thiol triggered hyperpolarized [128] Xe magnetofluorescent biosensor (**Probe 61**, Figure 54) based on a disulfide as the thiol recognition group. The probe composed of a TPP group as mitochondrial targeting ligand, a Xe encapsulated in a cryptophane-A cage as a [128] Xe NMR reporter, and a naphthalimide moiety as a fluorescent reporter. After the disulfide bond being cleavage by thiols, a cyclization reaction will occur to afford an amine product, resulting in a remarkable increase of fluorescence intensity at 560 nm and a significant change in the [128] Xe chemical shift. The probe exhibits an extremely low detection limit (10^−10^ M) by using Hyper-CEST (chemical exchange saturation transfer) NMR. The probe was successfully applied to effectively detect mitochondrial thiol changes in live cells.

#### 3.1.5. Detection of Mitochondrial NPSH through Other Methods

In 2014, a mitochondrion-targeting NIR probe (**Probe 62**, Figure 55) named MitoGP was developed by Soo-Yeon Lim et al. [179] for imaging and detecting mitochondrial thiols. Within the molecule, a heptamethine group was utilized as the mitochondrial targeting moiety and also served as the fluorophore, while a nitroazo group was used as the GSH recognition group as well as the fluorescence quencher. The probe showed a great selectivity for GSH over other amino acids including Cys and Hcy. The probe can be turned on to a strong fluorescence in the presences of GSH at 810 nm. The selectivity for GSH over Cys and Hcy was achieved by a specific 1, 6-conjugate addition and subsequent elimination reaction which triggered the release of the fluorophore [136]. In vitro cell experiments with HeLa cells demonstrated that the probe could effectively reflect the GSH level change in mitochondria. The authors believe that the MitoGP is much more superior to the commercially available thiol probe (mCB) or GSH-silent (rhodamine 123) MitoTracker and anticipated the possibility that the probe can be further used as a therapeutic reagent for mitochondrial GSH-related pathology. Moreover, the NIR emissive fluorescence property provides the advantage of avoiding the interference from cellular autofluorescence when used as noninvasive in vivo imaging tool.

In 2016, Fangfang Meng et al. [180] presented a two-photon fluorescent probe (**Probe 63**, Figure 55) for sensing mitochondrial thiols in live cells. Its two-photon properties enable the probe to be applied in mouse liver tissues. The probe named CA-TPP was developed by a combination of three parts, 9-ethyl-3-styryl-9*H*-carbazole (CA) as the TP fluorescent platform, aldehyde as the thiol recognition units, and a TPP as the mitochondrial targeting moiety. In vitro cellular experiments in A549 cells shows that pretreating the cells with NEM to deplete the mitochondrial thiol could lead to a decrease in fluorescence intensity by ~64%, indicating CA-TPP is able to reflect the mitochondrial thiols in live cells. CA-TPP was successfully applied for monitoring mitochondrial thiols in living tissues as well.

Hua Zhang et al. [181] developed an ultrasensitive ratiometric fluorescent probe (IQDC-M, **Probe 64**, Figure 56) for detection of ultratrace change of mitochondrial GSH in cancer cells with a detection limit to be as low as 2.02 nM. IQDC-M was designed by utilizing a modified sulfonamide as a thiol response moiety. Two naphthalene derivative fluorophores served as the fluorescence reporting groups are linked together by sulfonamide to maintain a fluorescence resonance energy transfer (FRET) process. In the presence of FRET, IQDC-M displays a maximum emission at 592 nm (Ex. 450 nm). Interestingly, IQDC-M show no fluorescence response when GSH changed at mM level due to the formation of N=N” group [182], while ultratrace change of GSH at nM could forbad the FRET and led to a significant fluorescence decrease at 592 nm and appearance of a new peak mainly at 520 nm from the product IQ-M which was a result of a thiol attack at the sulfonamide bond (addition reaction) followed by an elimination reaction to release IQ-M. The ratio of I_520 nm_/I_592 nm_ increased linearly as the GSH concentration increases in nM scale. The probe showed a high selectivity for GSH than other thiols, such as Cys, Hcy, or DTT. The probe was proved to be well located inside mitochondria via a colocalization study with MitoTrackers Green FM. Interestingly, the probe was found to selectively enter cancer cells. The compound has been successfully applied to monitor the ultrachange of mitochondrial GSH in cancer cells during the apoptosis for the first time.

In 2020, Pingru Su et al. [135] reported a TAT peptide-based ratiometric two-photon fluorescent probe (TAT-probe, **Probe 65**, Figure 57) which can detect thiols and simultaneously distinguish GSH in mitochondria. Cell penetrating peptide TAT (RRQRRKKRG) was widely used for delivering macromolecular substances into cells and mitochondria [183,184,185]. Thus, TAT was incorporated in the probe for achieving the mitochondria selectivity. In the meantime, a naphthalimide and a rhodamine B fluorophore were connected through a thioester bond to construct a Forster resonance energy transfer (FRET) system. A reaction with thiols breaks the thioester and releases a free green fluorescent naphthalimide derivative (Ex. 404/820 nm, Em. 520 nm) and a thiol-conjugated rhodamine. The GSH conjugated rhodamine displays a red fluorescence (Ex. 545 nm, Em. 585 nm) and Hcy/Cys conjugated rhodamine exhibit no fluorescence. TAT-probe was demonstrated to be able to detect GSH, Cys and Hcy with the detect limits of 5.15 μM, 0.865 μM, and 6.51 μM, respectively. Cellular experiments with HeLa cells showed that TAT-probe was able to detect thiols as well as discriminatively detect GSH over Cys/Hcy in mitochondria in live cells in different excitation channels.

In 2017, Chunlong Sun et al. [186] reported a mitochondria-targeted two-photon fluorescent probe (TP-Se, **Probe 66**, Figure 58) to monitor changes of ONOO^−^/GSH levels in cells based on a organoselenium moiety serving as the rection site for ONOO^−^/GSH and a methyl pyridinium moiety as a mitochondria-targeting functional group. The emission intensity at 565 nm (Ex. 430 nm) linearly increased upon reaction with ONOO^−^, while the fluorescence intensity increase was reversed when GSH was added. Cellular experiments revealed that TP-Se exhibited a good sensitivity and selectivity in monitoring ONOO^−^oxidation and GSH reduction events under physiological conditions in live cells.

In 2004, George T. Hanson et al. [187] reported redox-sensitive green fluorescence protein indicators (roGFPs, **Probe 67**) for monitoring mitochondrial redox potential. RoGFPs were constructed by introducing of redox-reactive groups (disulfides) to green fluorescence protein through gene modification. The leader sequence of the E1_ subunit of pyruvate dehydrogenase was employed to direct the protein to mitochondria. The probe exhibits two maximum wavelength bands of around 400 and 490 nm while monitoring emission at 508 nm. The ratio of fluorescence intensity of two wavelengths responds to the changes in ambient redox potential rapidly, reversibly, and ratiometrically. RoGFPs were proved to be able to monitor the mitochondrial redox potential changes caused by cell treatment with a reductants (DTT) or an oxidants (H_2_O_2_) in these cells through fluorescence microscopy or in cell suspension using a fluorometer.

### 3.2. Selective Detection of Mitochondrial Cys/Hcy

#### 3.2.1. Selective Detection of Mitochondrial Cys and Hcy through Cyclization of Cys/Hcy with Acrylates or Aldehydes

In 2015, Chunmiao Han et al. [188] presented a near-infrared mitochondria-targeting fluorescent probe (NFL_1_, **Probe 68**, Figure 59) for selective detection of Cys over Hcy and GSH with a low detection limit of 14.5 nM. NFL_1_ is composed of a semiheptamethine derivate group to serve as the NIR fluorophore and an acryloyl group to serve as a thiol responsive moiety. The fluorescence of the semiheptamethine derivate group was quenched by acryloyl group due to a PET process. The semiheptamethine derivate group also provides the probe the ability to target mitochondria due to a positive charge. The off-on mechanism triggered by Cys can be explained in two steps. First, a Michael addition of the thiolate from Cys to the acryloyl group to yield a thioether. Second, the thioether will go through a rapid intramolecular cyclization to form a seven-membered ring and release a free semiheptamethine derivate. This destroys the PET process and restores the strong NIR fluorescence of the semiheptamethine derivate with the maximum emission at 735 nm. The high selectivity of NFL_1_ toward Cys over other thiols is due to the fact that the kinetic rate of the intramolecular cyclization reaction with Cys is higher (formation of a seven-membered ring) than Hcy (formation of an eight-membered ring). The intramolecular cyclization reaction does not occur for GSH products. The fluorescence intensity at 735 nm is linearly increased as the concentrations of Cys ascended from 5 to 15 μM with a detection limit of 14.5 nM, enabling the quantitative detection of Cys. The probe has been proved to successfully target mitochondria and monitor mitochondrial Cys change. The reagent is also capable of assessing mitochondrial oxidative stress that induced by LPS in cells, Moreover, NFL1 was successfully applied in vivo for the detection of mitochondrial Cys.

In 2016, Weifen Niu et al. [112] developed a two-photon fluorescent probe (**Probe 69**) for ratiometrically imaging and detection of Cys in mitochondria in live cells. As illustrated in Figure 60, the probe named ASMI is made by attaching an acrylate moiety to a highly two-photon active and biocompatible merocyanine fluorophore, in which the acrylate moiety is employed as a thiol responsive site and merocyanine as the mitochondria targeting group since it carries a positive charge. Same as other acrylate probes, the reaction between Cys and the probe started with a Michael addition of Cys to the acrylate followed by an intramolecular cyclization to release the merocyanine fluorophore and a seven membered-ring cyclic amide. The probe ASMI itself showed blue fluorescence with a maximum emission wavelength at 452 nm while the released free merocyanine exhibited green fluorescence with a maximum emission wavelength at 452 nm. In addition, both ASMI and merocyanine showed large two-photon action cross-section (Φσ_max_) of 65.2 GM (Ex. 740 nm) and 72.6 GM (Ex. = 760 nm), respectively. In addition, the ratio of fluorescence intensity of emission at 518 nm and 452 nm (F_518_/F_452_) is linearly proportional to Cys concentrations in the range of 0.5−40 μM, suggesting that the probe can detect Cys’ level ratiometrically. All of these features raise the high potential to use the probe for high contrast and brightness ratiometric two-photon fluorescence imaging of live samples. Compared to other thiols such as Hcy or GSH, ASMI reacted much faster with Cys. Moreover, the probe was able to ratiometrically detect the mitochondrial Cys change in live cells when its level is elevated by addition of Cys or decreased by NEM. ASMI was also demonstrated to selectively detect mitochondrial Cys and monitor Cys status change in intact living tissues at the depth of 150 μm by the two-photon fluorescence microscopy in a mouse model.

In addition, a number of other probes have been developed by using the same sensing mechanism. For instance, Chae Yeong Kim et al. [189] developed a mitochondrial Cys imaging probe by caged oxazolidinoindocyanine, Peng Zhang et al. [190] developed a NIR probe based on a difluoroboron curcuminoid scaffold, and LijunTang et al. [191] reported a far-red emissive fluorescence probe based on benzothiazole, etc.

#### 3.2.2. Selective Detection of Mitochondrial Cys/Hcy through Cys-induced SNAr Substitution−Rearrangement Reaction

Mingwang Yang et al. [192] reported a colorimetric and ratiometric fluorescent chemosensor (**Probe 70**, Figure 61) for the detection of Cys/Hcy in mitochondria with a high selectivity and sensitivity (Cys, 22 nM; Hcy, 23 nM) in 2019. The probe is made by connecting a 4-methlythiophenol to a positive charged pyronin fluorophore with a thioether bond which also serves as the thiol recognition unit. The selectivity of Cys/Hcy over GSH is achieved by a SNAr substitution reaction followed by an intramolecular rearrangement reaction while GSH is not able to undergo the intramolecular rearrangement reaction. Upon reaction with Cys/Hcy, fluorescence intensity ratio of F_540_/F_620_ increases linearly with the concentration of Cys/Hcy, enabling the raitometric detection of Cys/Hcy. The probe was not only able to reflect mitochondrial thiol change in MCF-7 cells induced by H_2_O_2_, but also able to ratiometrically image endogenous and exogenous thiols in living organisms.

In 2021, Xin Ji et al. [193] reported a pyridinium substituted BODIPY probe ((BDP-S-*o*-Py^+^, **Probe 71**, Figure 62) for selective and rapid detection of mitochondrial Cys in vitro and in vivo based on the SNAr substitution−rearrangement reaction. The pyridinium group was connected to a BODIPY fluorophore via a thioether bond to serve as a thiol recognition group and a mitochondrion targeting moiety. As a good leaving group, the pyridinium will be replaced by the thiolate of Cys upon reaction. Then an intramolecular rearrangement reaction occurred to produce an amine-substituted BODIPY with significantly different fluorescence properties. The detection limit of the compound was determined to be 72 nM with a linear relationship between the fluorescence intensity and Cys concentration in a range of 0 to 50 μM. The probe was successfully used to detect mitochondrial Cys change in HeLa cells with minimum interference from GSH and Hcy. The probe was capable of monitoring endogenous Cys in mice as well.

#### 3.2.3. Selective Detection of Mitochondrial Cys/Hcy through Other Mechanisms

In 2019, Li Fan et al. [194] reported the first mitochondria-targeted ratiometric two-photon (DNEPI) probe (**Probe 72**, Figure 63) based on the DNBS group. The probe is prepared by combining the two-photon fluorophore, merocyanine, and a thiol reaction unit DNBS, while a positive charge of the structure was accounted for mitochondria targeting. A cleavage of the sulfonic acid ester bond by Cys leads to a significant red shift of both fluorescence (485 to 583 nm) and absorption properties (352 to 392 nm). The ratio of fluorescence intensity (F_583_ nm/F_485_ nm) displays a linear relationship with Cys concentration (2–10 μM), and the detection limit was determined to be 0.29 μM. A much higher reactivity of Cys than GSH and Hcy leads to the selectivity of the probe for Cys. The probe was capable of monitoring the level of intracellular Cys.

Kun Yin et al. [195] reported in 2015 a near-infrared ratiometric fluorescence probe Cy-NB (**Probe 73**) for detection of mitochondrial Cys. The probe is composed of heptamethine cyanine as fluorophore and *p*-nitrobenzoyl as the Cys recognition site. Heptamethine cyanine also serves as the mitochondria targeting moiety (Figure 64). The fluorescence properties of heptamethine cyanine are changed dramatically when *p*-nitrobenzoyl is attached to modulate the intramolecular polymethine π-electron system. Cleavage of *p*-nitrobenzoyl from the probe by Cys will restore the polymethine π-electron system and induce a remarkable fluorescence emission shift simultaneously. The probe itself shows a maximum excitation wavelength and maximum emission wavelength of 720 nm and 785 nm, respectively. Upon reaction with thiols, the maximum excitation and emission wavelengths change to 580 nm and 640 nm due to the released heptamethine cyanine. The emission peak at 785 nm (Ex. 720 nm) of probe Cy-NB decreases gradually while the emission peak at 640 nm (Ex. 560 nm) increases gradually in response to an increase in Cys concentration. The ratio of two peaks (F_640_ nm/F_785_ nm) followed a linearly change when the concentration of cysteine is in the range of 0–35 μM and 35–100 μM, and the detection limit is 0.2 μM in 5 min. The lower pKa of Cys (8.53) than Hcy (10.00) and GSH (9.20) is believed to be one of the reasons attributing to the Cys selectivity because more nucleophilic ionized form will be formed for Cys. Moreover, the steric hindrance of tripeptide GSH is believed to be another reason for the selectivity. The pseudo-first-order reaction constant *k* for Cys is 330-fold higher than GSH and 78-fold higher than Hcy. The probe was applied for detection of mitochondrial Cys’ status in HepG2 cells successfully. The probe can reflect the Cys level change in mitochondria. The probe and is capable of detecting endogenous Cys level in mice.

In addition to the probes developed above, many other probes have been developed as well for selective detection of mitochondrial Cys or Hcy., Xiaopeng Yang et al. [196] developed a multi-signal fluorescent probe for selective detection of Cys and SO_2_ based on a coumarin fluorophore. Jing Liu et al. [197] developed a highly sensitive mitochondrial Cys detection probe based on a coumarin-hemicyanine fluorophore.

### 3.3. Lysosomal Total Thiols and GSH

As another important subcellular organelle, lysosomes serve as the main digestive compartments with more than 50 hydrolases for digesting various exogenous and endogenous biomolecules. Thiols help with the processing of macromolecule degradation by the reduction of disulfide bonds [198,199,200,201]. Additionally, GSH has been reported to stabilize lysosome membranes while Cys is known for activating albumin degradation in liver lysosomes in mice [202]. Therefore, the importance of assessment of thiol levels in lysosomes has been well recognized. However, not many probes have been reported for detecting thiols in lysosomes in live cells.

The design of most of lysosome targeting thiol probes involves the incorporation of a weak basic group which can be trapped in lysosomes which are acidic (pH ~5.0). 4-(2-Aminoethyl)morpholine and *N*,*N*-dimethylethylenediamine are the most widely used base molecules for lysosomes targeting [26,136] (Figure 65).

#### 3.3.1. Detection of Lysosomal NPSH via a Michael Addition Reaction

In 2016, Rong Huang et al. [203] reported a lysosome targeting rhodamine B-based fluorescent probe (**Probe 74**, Figure 66) for rapid and sensitive detection of GSH utilizing a Michael acceptor as the thiol response group. The probe itself is non-fluorescent due to PET but can be turned on to fluorescent (Ex. 520 nm, Em. 582 nm) by GSH within in 10 s. The “turn on” can be explained by a Michael addition reaction followed by the formation of an H-bond between the rhodamine’s carbonyl group and the *N*-H group of GSH to open the ring of rhodamine resulting in a remarkable fluorescence increase. The weak basicity of the probe attributes to the lysosome selectivity. The probe is featured with a high sensitivity with a detection limit of 190 nM. Cellular experiments and colocalization studies showed that the probe effectively accumulated inside lysosomes and could detect the GSH level change in lysosomes in HeLa cells.

#### 3.3.2. Detection of Lysosomal NPSH via SNAr Reactions Using Ether, Thioether as a Leaving Group

##### Detection of Lysosomal NPSH via SNAr Reactions Using Ether as a Leaving Group

In 2018, Hui Zhang et al. [204] reported a lysosome targeting, coumarin and resorufin based fluorescent probe (**Probe 75**, Figure 67) for simultaneous detection of Cys/Hcy and GSH. A morpholine moiety was introduced to serve as the lysosome targeting group, and the coumarin derivative and resorufin were connected together by an ether bond which was the thiol response site. Reaction with different thiols by a nucleophilic substitution reaction will lead to the same free red fluorescent resorufin but different thio-coumarin. The thio-coumarin of Cys/Hcy undergoes an intramolecular rearrangement to switch from thio-coumarin to amino-coumarin (blue) with a significantly fluorescence change while the thio-coumarin (green) derived from GSH will remain unchanged. Thus, the probe can be used to detect the total thiols by measuring the red fluorescence of red fluorescent resorufin and detect Cys/Hcy, and GSH simultaneously from their corresponding coumarin adducts. A nearly linear relationship was found between the fluorescence intensity at 480 nm and Cys/Hcy concentration with a detection limit determining of 27 nM (or 33 nM for Hcy). A linear relationship and a detection limit of 16 nM were found for GSH with its corresponding fluorescence intensity at 542 nm. Cellular experiments with HeLa cells demonstrated that the probe was cable of discriminating intracellular Cys/Hcy, GSH based on their different signal patterns.

Xufen Song et al. [205] published their work in 2020 for the development of a lysosome-targeting fluorescent probe (**Probe 76**, Figure 68) for simultaneous detection and discrimination of Cys/Hcy and GSH by dual channels based on a similar mechanism. The probe named Lyso-O-NBD was constructed by combining a coumarin fluorophore and a NBD fluorophore via an ether bond, and a lysosome targeting moiety morpholine was attached to the Coumarin fluorophore. Similar with the probe discussed above, an intramolecular rearrangement after the nucleophilic substitution reaction attributed to the fluorescence difference of the GSH adduct and the Cys/Hcy adducts. A linear relationship was also found between the thiol (GSH, Cys, or Hcy) concentration and fluorescence intensity at 490 nm with a detection limit of 3.3 × 10^−8^ M, 5.2 × 10^−8^ M, and 3.9 × 10^−8^ M for Cys, Hcy, and GSH, respectively. Later, Lyso-O-NBD was successfully applied to detect and discriminate Cys/Hcy and GSH in lysosomes in HeLa cells by confocal laser scanning microscopy.

##### Detection of Lysosomal NPSH via SNAr Reactions Using Thioether as a Leaving Group

Based on the thiol specific thiol–sulfide exchange reaction reported from our group [42], we developed a thiol specific and lysosome-selective fluorogenic agent (BISMORX, **Probe 77**, Figure 69) [206] in 2019 by introducing a lysosome targeting structure morpholine symmetrically to the symmetric benzofurazan sulfide. BSMORX uses a thioether moiety to serve as the thiol recognition units. Similar to the early thiol specific fluorescence imaging probes developed from our lab, BISMORX itself is non-fluorescent due to the self-quenching of the symmetric structure but reacts readily with a thiol to form a strong fluorescence thiol adducts (Ex. 380 nm, Em. 540 nm). Cellular experiments showed that BISMORX was able to image, quantify, and detect the change of NPSH in lysosomes in live cells. We also developed another lysosomal NPSH imaging agent TBONES (**Probe 78**) [207]. Interestingly, the lysosome selectivity of this probe was likely due to its selective entry of lysosomes through an endocytosis pathway which is different than other lysosome-targeting probes.

#### 3.3.3. Detection of Lysosome NPSH via Cleavage of Sulfonamide and Sulfonate

##### Detection of Lysosomal NPSH via Cleavage of Sulfonamide

In 2016, Meijiao Cao et al. [208] reported a lysosome targeting naphthalimide-based fluorescence probe (**Probe 79**, Figure 70) for selective detection of GSH in live cells based on cleavage of sulfonamide as the thiol recognition mechanism. A morpholine was attached to the probe to provide the lysosome selectivity. Cleavage of sulfonamide by GSH will produce a strong fluorescent GSH conjugated naphthalimide derivate (Ex. 370 nm, Em. 495 nm). Colocalization experiments with LysoTracker Red on HepG2 cells demonstrated that the probe effectively localized inside lysosomes. Further cellular experiments showed that the probe effectively reflected lysosome GSH changes but not affected by a change in the levels of Cys and Hcy in HepG2 cells.

Jiangli Fan et al. [165] reported a similar work (**Probe 80**, Figure 71) in 2016 in which naphthalimide was used as the fluorophore. A DNBS group was attached to serve as a thiol recognition group and a morpholine was incorporated as the lysosome targeting ligand. PET between DNBS and naphthalimide quenches the fluorescence of the probe. Upon reaction with thiols, DNBS will be detached and terminate the PET process, leading to the restoration of the fluorescence of naphthalimide (Ex. 400 nm, Em. 540 nm). A linear relationship was found between the fluorescence intensity at 540nm and the concentration of thiols with a detection limit of 2.6 × 10^−7^ M, 2.41 × 10^−6^ M and 4.87 × 10^−6^ M for Cys, GSH, and Hcy, respectively. The probe was successfully used to detect lysosomal thiol change in live HeLa cells and MCF-7 cells as well as in tissues by a two-photon microscopy.

##### Detection of Lysosomal NPSH via Cleavage of Sulfonate

A lysosome targeting ruthenium(II) derivative probe (**Probe 81**, Figure 72) was developed by Quankun Gao et al. [209] in 2017 for selective detection of thiols based on the cleavage of sulfonate mechanism. A morpholine moiety was incorporated into the complex to achieve the lysosome selectivity, DNBS was attached to serve as an efficient PET quencher. Upon reaction with thiols, a phosphorescence enhancement was observed at 620 nm (Ex. 459 nm) due to the cleavage of sulfonate moiety. A linear relationship was observed between the phosphorescence intensity and the concentration of GSH in a range of 4.0 to 25 μM. The detection limit was determined to be 62 nM, 146 nM, and 115 nM for GSH, Cys, and Hcy, respectively. The probe was successfully applied for visualizations of thiols in lysosomes in live cells and in *Daphnia magna*.

#### 3.3.4. Detection of Lysosomal NPSH via Cleavage of Disulfide

In 2019, Ziming Zheng et al. [210] reported a disulfide bond based probe (SQSS, **Probe 82**) for selective detection of lysosome thiols in cells with a high signal-to-noise ratio. Two squaraines (NiR fluorophore) were linked together via a disulfide bond and a high fluorescence resonance energy transfer (FRET) quenching effect was generated between the two squaraines. The two weakly alkaline groups in the structure might attribute to the lysosome selectivity. Upon reaction with GSH, the double bond will be broken leading to the restoration of the fluorescence (Em. 665 nm, Ex. 610 nm). A linear relationship was observed between the concentration of GSH and fluorescence intensity (665 nm) with a detection limit of 0.15 μM, and the probe showed a higher selectivity toward GSH than Cys and Hcy by a possible electrostatic interaction between the two carbonate anions of GSH and the thiazole cations of SQSS. Cell imaging experiments revealed that SQSS can monitor endogenous and exogenous GSH in tumor or normal cells.

#### 3.3.5. Detection of Lysosome NPSH via Other Methods

In 2020, Hong Wang et al. [211] built a dual targeting (cancer-specific and lysosome-targeted) fluorescence nanoprobe (DTFN, **Probe 83**) for GSH imaging in live cells. The nanoprobe was constructed by combining folic acid-modified photostable aggregation-induced emission dots with manganese dioxide (MnO_2_) nanosheets (GSH responsive site) through electrostatic interactions. Folic acid was introduced to facilitate the probe to be taken by folate receptor (FR) over-expressed cancer cells, and the positively charged amino moiety helps the nanoparticles enter lysosomes. Fluorescence quenching is achieved vis a fluorescence resonance energy transfer (FRET) effect from AIE dots to the MnO_2_ nanosheets. Reaction with GSH will reduce MnO_2_ nanosheets to Mn^2+^ to break the FRET process resulting in the restoration of the fluorescence at 523 nm. The fluorescence intensity increased linearly as the concentration of GSH increased with a detection limit of 1.03 μM. Cellular experiments with live HeLa cells showed that the probe could successfully detect lysosomal GSH level in cancer cells.

### 3.4. Selective Detection of Lysosomal Cys and Hcy

Development of probes for detection of lysosomal Cys and Hcy are still in the early stage. In 2019, Jinhua Gao et al. [212] reported two lysosome targeting BODIPY fluorescence probes (**Probes 84–85**) for selective detection of Cys over GSH and Hcy based on a Cys induced SNAr substitution reaction followed by an intramolecular rearrangement. As illustrated in Figure 73, morpholinoethoxy group was introduced to achieve the lysosome selectivity, and a *p*-methoxyphenylmercapto moiety was attached to the meta position to serve as a Cys recognition unit. After reaction with Cys, the *p*-methoxyphenylmercapto moiety was replaced by Cys by a Cys-induced SNAr substitution followed by an intramolecular rearrangement to generate meso-amino-BODIPYs with strong fluorescence at 566 nm. The two probes displayed high selectivity for Cys over GSH and Hcy. A linear relationship was found between the fluorescence intensity (566 nm) and Cys concentration and a low detection limit was achieved for Cys (46 nM for Lyso-S and 76 nM for Lyso-D). Cellular experiments in HeLa cells showed that the probes effective localized in lysosomes and imaged lysosome thiols.

### 3.5. Detection of NPSH in Endoplasmic Reticulum

Endoplasmic reticulum plays important roles [136] in protein synthesis, lipid and carbohydrate metabolism, calcium signaling, etc. Fluorescent probes for imaging thiols inside endoplasmic reticulum have been rarely reported. Among a very few reported probes, methyl benzene sulfonamide is the most commonly used as an ER-targeting moiety (Figure 74) [115,213,214].

In 2019, Chengshi Jiang et al. [215] developed an naphthalimide-based fluorescent probe (**Probe 86**, Figure 75) for imaging cellular GSH in the ER in live cells. The probe named ER-G was constructed by introducing a *p*-toluenesulfonamide unit to the top of naphthalimide as an ER-targeting unit and a DNBS was attached to the tail as the thiol recognition unit. The non-fluorescent ER-G can be turned on to a highly fluorescent naphthalimide derivate (Em. 558 nm, Ex. 470 nm) after reaction with GSH in which the DNBS was cleaved. The fluorescence intensity was found to be linearly associated with the concentration of GSH (75–300 μM). Cellular experiments with live HepG2 cells revealed that the probe detected the thiol level change by addition of NEM or addition of GSH. The probe was capable of monitoring H_2_O_2_-induced cellular GSH concentration change in the ER.

A ER-targeted TP fluorescent probe named as ER-SH (**Probe 87**, Figure 75) based the mechanism of cleavage of sulfonamide has been developed for imaging of thiols in live cells and in vivo by Ping Li et al. in 2019 [214]. The probe was constructed similarly with the probe mentioned above in which a naphthalimide was used as a two-photon fluorophore, a methyl benzene sulfonamide was used as an ER targeting moiety and DNBS was used as a thiol recognition group. ER-SH exhibits minimum fluorescence since the fluorescence of naphthalimide was quenched by DNBS due to a PET mechanism. Reaction with thiol will cleave the DNBS and restore the fluorescence (535 nm). Fluorescence intensity at 535 nm increased linearly with an increase in the concentrations of Cys, GSH, and Hcy with the detection limit of 1.67 × 10^−7^ M, 4.70 × 10^−6^ M, and 9.62 × 10^−7^ M, respectively. In addition to the ability to reflect thiol levels’ change in live cells, the probe was demonstrated to measure thiols levels in vivo by revealing that the thiol levels were reduced in brains of mice with depression phenotypes.

Xiuxiu Yue et al. [216] developed an ER selective fluorescent probe (ER-CP, **Probe 88**, Figure 76) to discriminatively detect Cys, Hcy, and GSH in live cells for the first time based on a SNAr substitution reaction followed by an intramolecular rearrangement reaction. ER-CP composed of three parts, a coumarin fluorophore, a methyl sulfonamide as an ER-targeting group, and 4-nitrobenzenethiol attached to the coumarin through a thioether bond as the thiol recognition group. PET presented between 4-nitrobenzenethiol and coumarin leads to a fluorescence quenching of ER-CP. A reaction with thiols will replace the nitrobenzenethiol moiety to form a green fluorescent thiol adducts, the same as other probes we described before. The thiol adducts with coumarin derived from Cys and Hcy will further undergo an intramolecular rearrangement reaction to afford a blue fluorescent amnio-substitution coumarin adducts (Ex. 370 nm, Em. 473 nm) although the reaction for Cys was much fast than Hcy while GSH adducts will remain unchanged as the green fluorescent thiol-substitution adducts (Ex. 427 nm, Em. 541 nm). A linear relationship was found between the concentration of thiols (Cys, Hcy, and GSH) and its corresponding fluorescence intensity with a detection limit of 14 nM, 16 nM, and 23 nM, respectively. The probe demonstrates an excellent ER-targeting property and is able to simultaneously detect Cys, Hcy, and GSH in HeLa cells.

## 4. Conclusions

In the past two decades, extensive efforts have been made in thiol detection in live cells with different approaches and mechanisms. Numerous fluorescent probes have been developed for detection of thiols in the whole cell as well as in subcellular organelles. The probes for imaging thiols in the whole cells are usually made up of two parts: a structure that can react with a thiol selectively, ideally specifically, and a structure that can be fluorescent for detection. The probes for imaging thiol in subcellular organelles will need to add a subcellular organelle targeting part on top of the probe for imaging thiols in the whole cells. The selectivity and reactivity of a probe for thiol is critical in ensuring only thiols are imaged. This is especially true for quantification. As discussed early, most of the employed reactions for thiol detection are based on a nucleophilic reaction. Although thiol is preferred in these reactions based on the fact that thiol is superior in terms of its nucleophilicity when compared with other common nucleophilic groups present in the biological system such as -OH, -NH, and COOH, these reactions are not thiol specific. A few reactions for thiol were reported to be thiol specific. These reactions include thiol–disulfide exchange reaction and thiol–sulfide exchange reaction (Section 2.1.4, Section 3.1.4, Section 3.3.4, Section 2.1.2.3, Section 3.1.2.2 and Section 3.3.2.2). It is expected that significant progress will be made if more thiol specific reactions are identified.

This review summarizes various thiol imaging probes used for live cell thiol imaging. To help readers for the probes presented in this review, Table 1 lists the mechanisms of thiol detection, usage, detection limits, and references for most of the probes presented in this review.

Developing probes for imaging thiols in subcellular organelles obviously faces more challenges since it requires the availability of the subcellular targeting part. Thiol probes for many subcellular organelles including nucleus, ER, Golgi apparatus, and cell surface are still lacking due to a lack of an ideal targeting ligand for the subcellular organelle. Although a number of probes have been developed for imaging thiols in mitochondria and lysosomes, challenges still exist. For example, the majority of mitochondria targeting probes achieved their mitochondria targeting by attaching a lipophilic cationic structure which has the potential risk of decreasing negative mitochondrial transmembrane potential (MMP) [136]. The lysosome targeting group morpholine can induce certain toxicities by elevating the pH levels of lysosomes if a high amount of the probes are applied [136].

In addition, a low water solubility is another shortcoming for most of the probes. Organic solvents like DMSO are commonly used to improve water solubility to enable the probe to be applied in cells or tissues. Thus, a probe with adequate water solubility and sufficient permeability is desirable [136]. A membrane transporter can also be a problem. Although most of the probes can diffuse inside cells due to a good permeability, active transporters (e.g., P-gp, or MRP) on cell membrane can potentially pump the hydrophobic probes out of the cells rapidly, which can significantly affect the accuracy of thiol measurement in live cells. The potential interferences caused by active transporters should be studied.

Real-time thiol imaging in live cells is another challenge. Most of the probes developed are based on an irreversible reaction with thiols, which makes it difficult to monitor the dynamic change of thiols in real time. A few reagents that can detect the analysts of interest in a reversible manner have been developed in the past ~5 years to address this challenge [74,75,76,77,139,140,217]. These probes can monitor the dynamic status change of the analysts in real time. Some of the reversible probes have been well developed and characterized for quantifying and monitoring GSH changes at the cellular level and even subcellular levels with a number of probes showing the capability of quantifying and monitoring mitochondrial GSH changes [139,140].

More agents for different subcellular organelles are expected to be developed, and better fluorophores and sensing mechanisms are also anticipated to be discovered and applied in the future. Although majority of the developed probes can only be used for cells, quite a number of probes with NIR or two-photon fluorophores have been successfully applied at tissue levels or in in vivo models. More impressively, some of the probes were demonstrated to be capable of distinguishing tumor tissues from other tissues, warranting the possibility of employing these probes as clinical diagnostic tools in the future though much work is still needed.

In conclusion, we summarized the design strategies and sensing mechanisms of some of the most representative thiol imaging probes for live cell thiol imaging. We hope this review can provide the readers an overview and better understanding of the current status and challenges in the field.

## Figures and Tables

**Figure 1 molecules-26-03575-f001:**
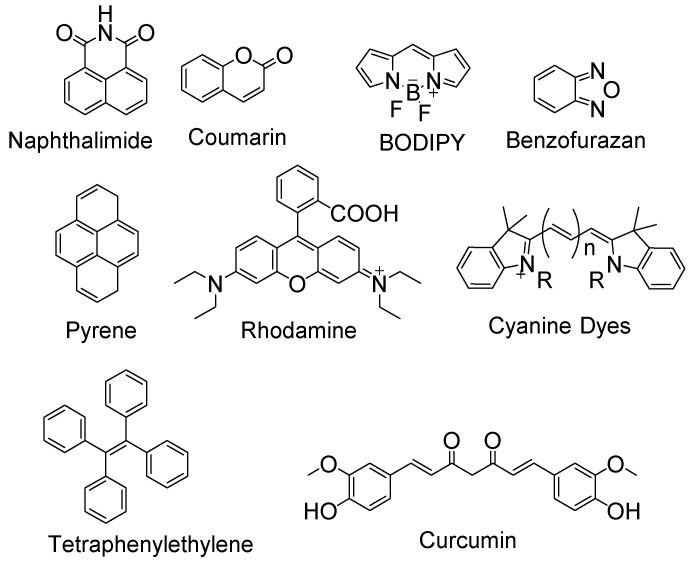
Different fluorophores used for construction of thiol detection probes.

**Figure 2 molecules-26-03575-f002:**
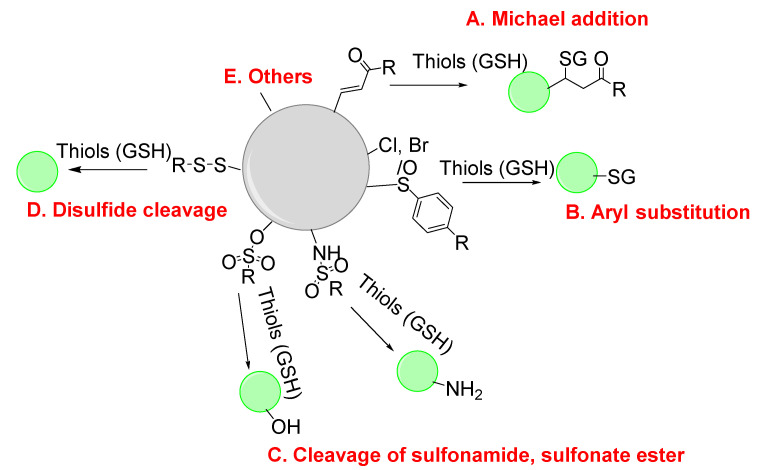
Representative sensing mechanisms used for thiol detection.

**Figure 3 molecules-26-03575-f003:**
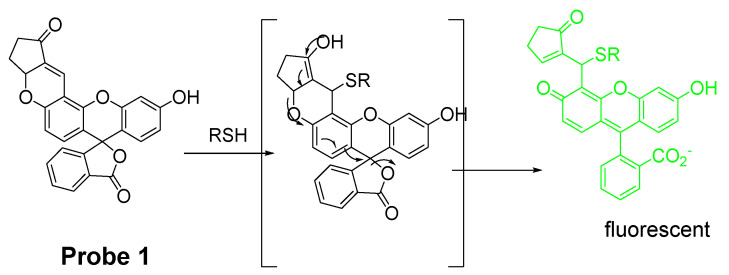
Structure of **Probe 1** and its reaction mechanism with RSH.

**Figure 4 molecules-26-03575-f004:**
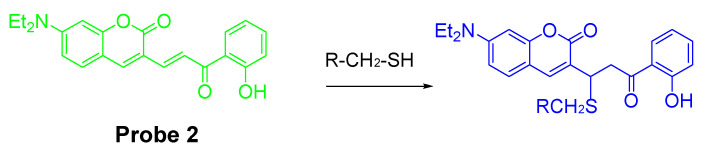
Structure of **Probe 2** and its reaction with RSH.

**Figure 5 molecules-26-03575-f005:**
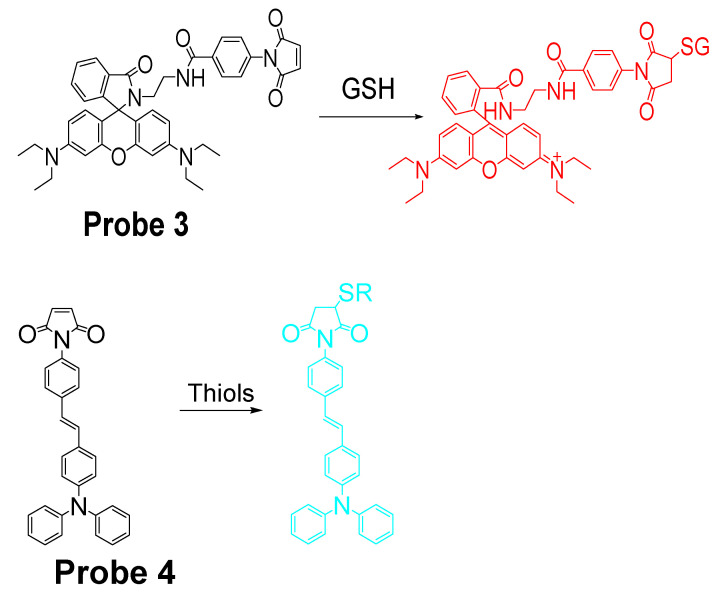
Structures of **Probes 3** and **4** and their reactions with GSH or thiols, respectively.

**Figure 6 molecules-26-03575-f006:**
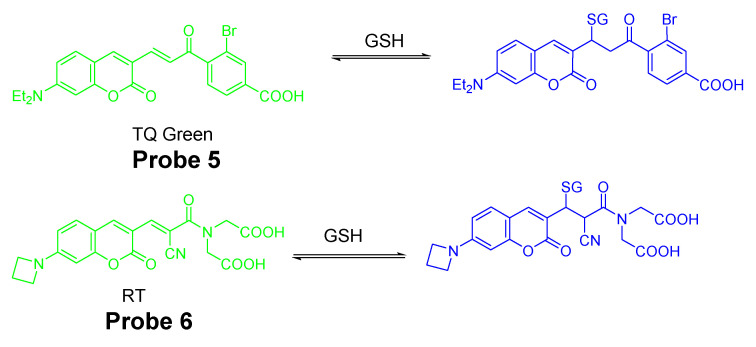
Structures of **Probes 5**, **6** and their reversible reactions with GSH.

**Figure 7 molecules-26-03575-f007:**
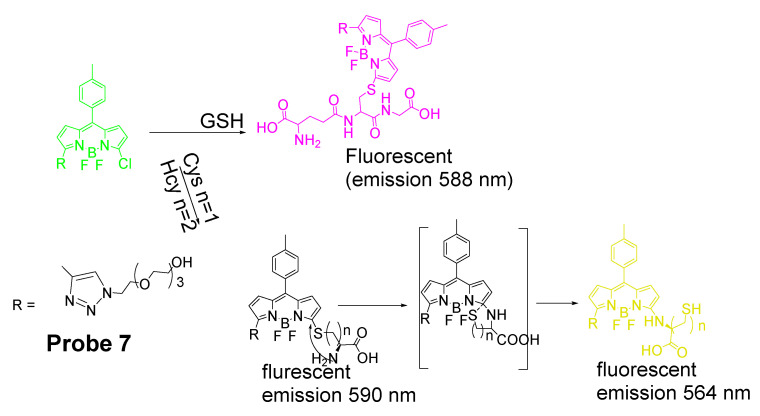
**Probe 7** and its reaction with GSH or Cys (*n* = 1)/Hcy (*n* = 2).

**Figure 8 molecules-26-03575-f008:**
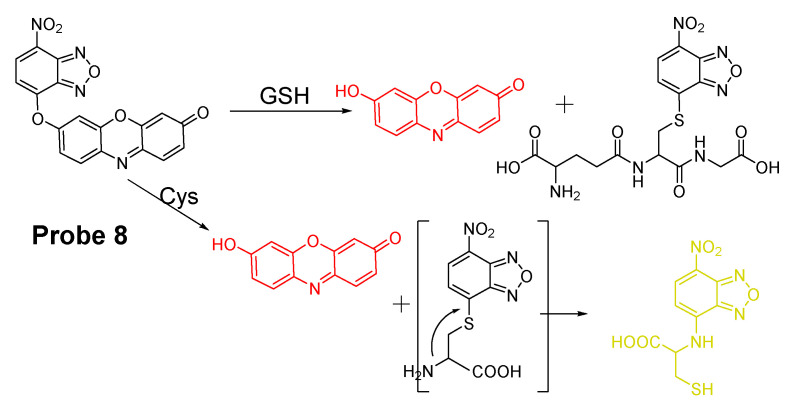
Structure of **Probe 8** and its reaction with GSH and Cys.

**Figure 9 molecules-26-03575-f009:**
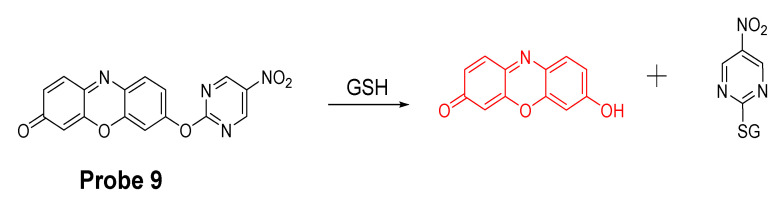
**Probe 9** and its reaction with GSH.

**Figure 10 molecules-26-03575-f010:**
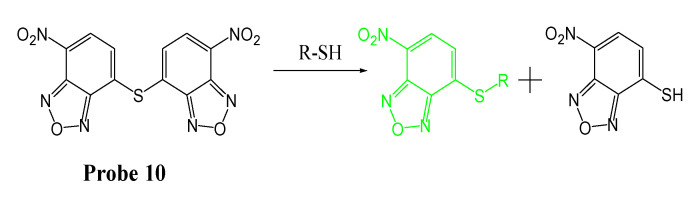
**Probe 10** and its reaction with thiols.

**Figure 11 molecules-26-03575-f011:**
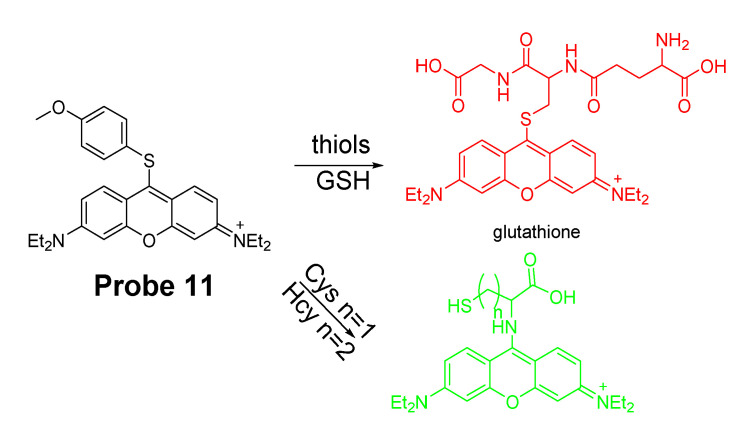
**Probe 11** and its reaction with GSH or Cys/Hcy.

**Figure 12 molecules-26-03575-f012:**
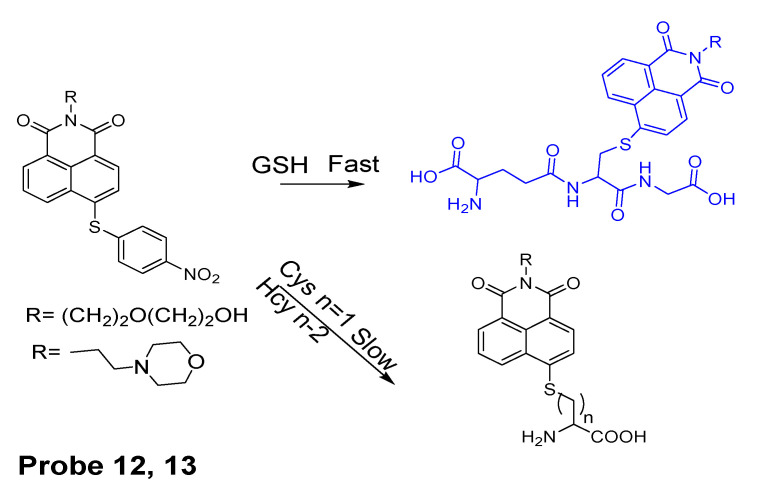
Structures of **Probes 12**, **13** and their reaction with GSH or Cys/Hcy.

**Figure 13 molecules-26-03575-f013:**
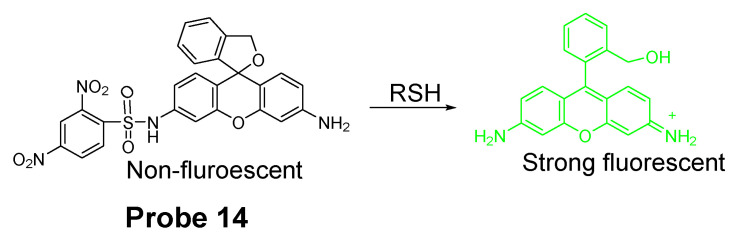
Structure of **Probe 14** and its reaction with thiols.

**Figure 14 molecules-26-03575-f014:**
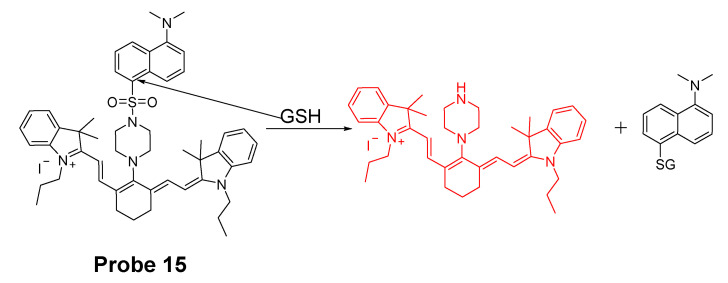
Structure of **Probe 15** and its reaction with GSH.

**Figure 15 molecules-26-03575-f015:**
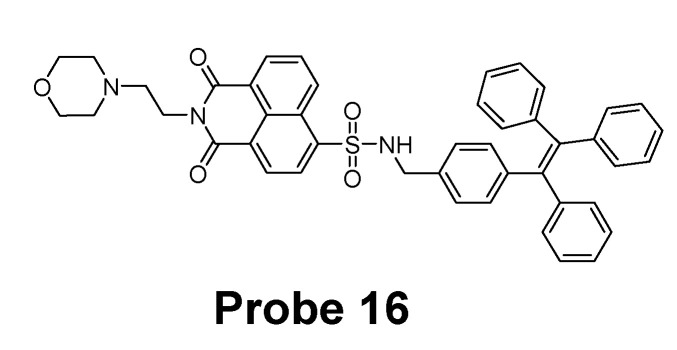
Structure of **Probe 16**.

**Figure 16 molecules-26-03575-f016:**
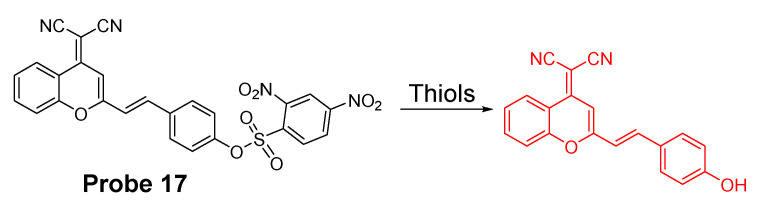
Structure of **Probe 17** and its reaction with thiols.

**Figure 17 molecules-26-03575-f017:**
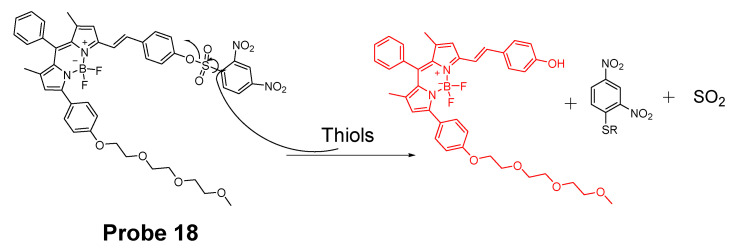
Structure of **Probe 18** and its reaction with thiols.

**Figure 18 molecules-26-03575-f018:**
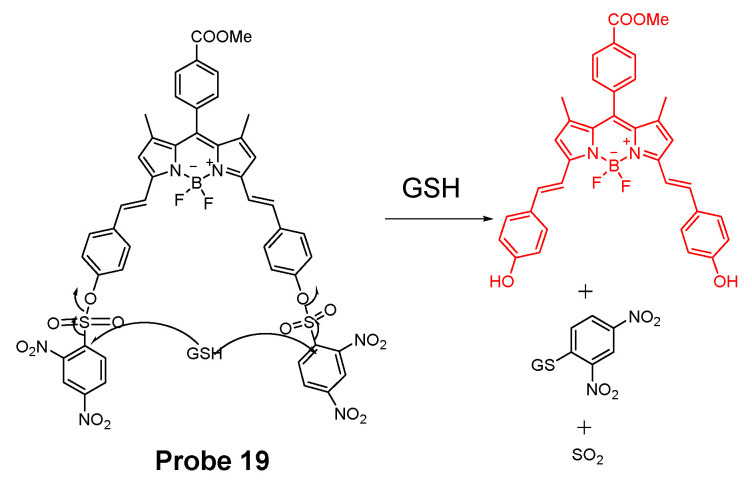
**Probe 19** and its reaction with thiols.

**Figure 19 molecules-26-03575-f019:**
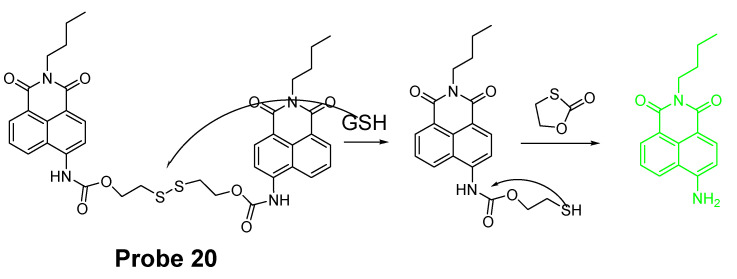
Structure of **Probe 20** and its reaction with GSH.

**Figure 20 molecules-26-03575-f020:**
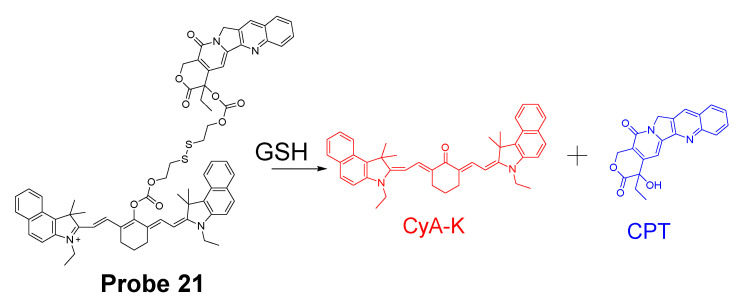
Structure of **Probe 21** and its reaction with GSH.

**Figure 21 molecules-26-03575-f021:**
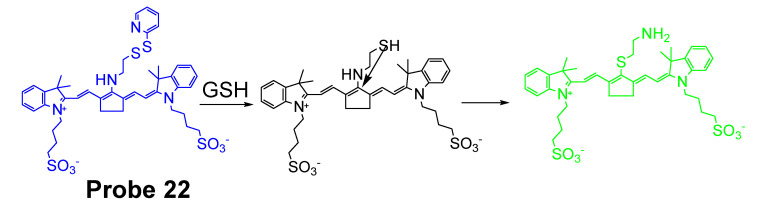
Structure of **Probe 22** and its reaction with GSH.

**Figure 22 molecules-26-03575-f022:**
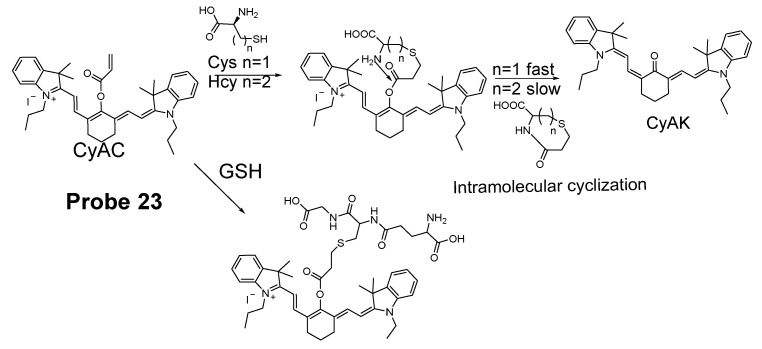
**Probe 23** and its reaction mechanism for distinguishing Cys over GSH and Hcy.

**Figure 23 molecules-26-03575-f023:**
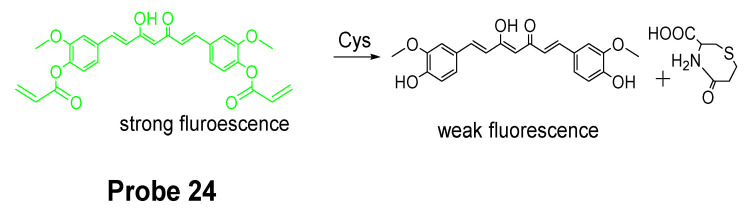
Structure of **Probe 24** and its detection mechanism with Cys.

**Figure 24 molecules-26-03575-f024:**
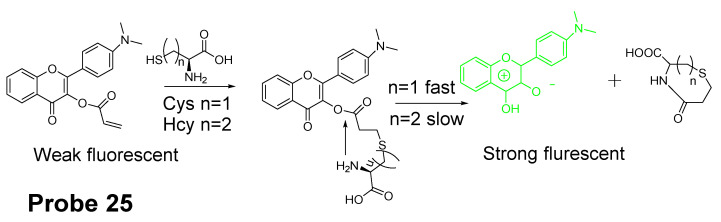
Structure of **Probe 25** and its reaction with Cys and Hcy.

**Figure 25 molecules-26-03575-f025:**
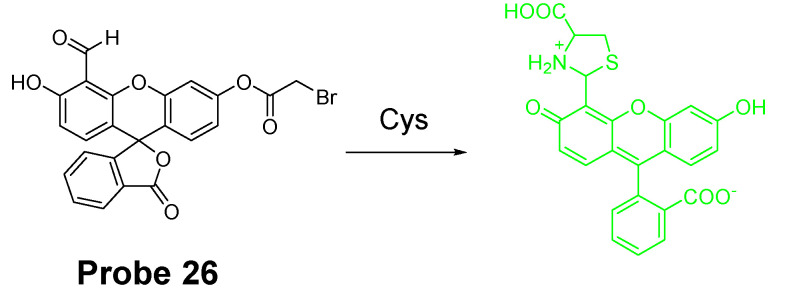
Structure of **Probe 26** and its reaction with Cys.

**Figure 26 molecules-26-03575-f026:**
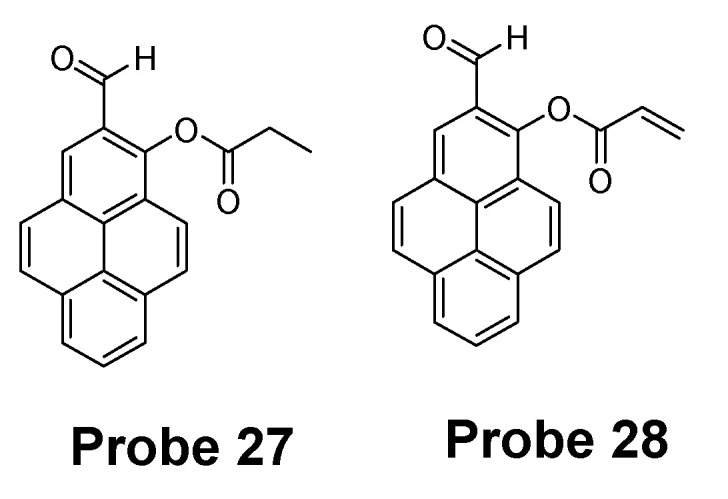
Structures of **Probes 27** and **28**.

**Figure 27 molecules-26-03575-f027:**
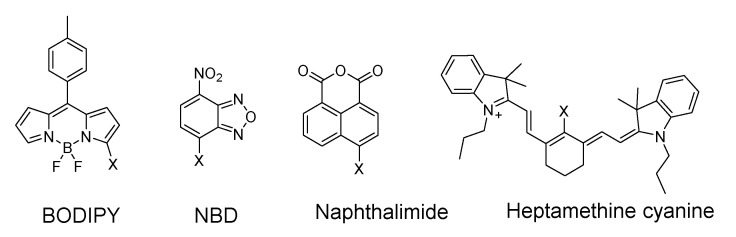
Representative fluorophores used for selective discrimination of thiols via Cys-induced SNAr substitution−rearrangement reaction.

**Figure 28 molecules-26-03575-f028:**
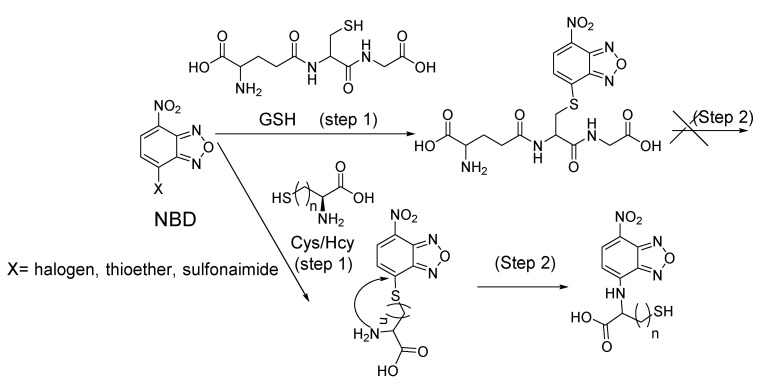
Illustration of the Cys-induced SNAr substitution−rearrangement reaction.

**Figure 29 molecules-26-03575-f029:**
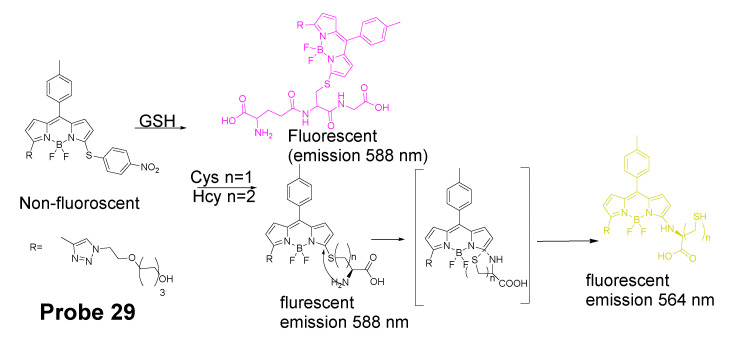
Structure of **Probe 29** and its sensing mechanisms for distinguish Cys/Hcy and GSH.

**Figure 30 molecules-26-03575-f030:**
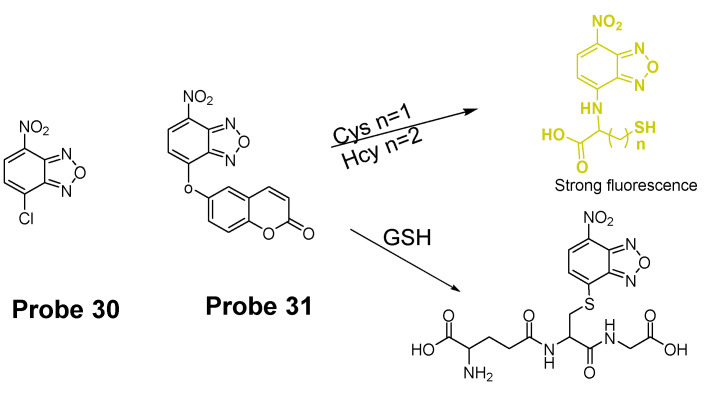
Structures of **Probes 30**, **31** and their reaction toward GSH and Cys/Hcy.

**Figure 31 molecules-26-03575-f031:**
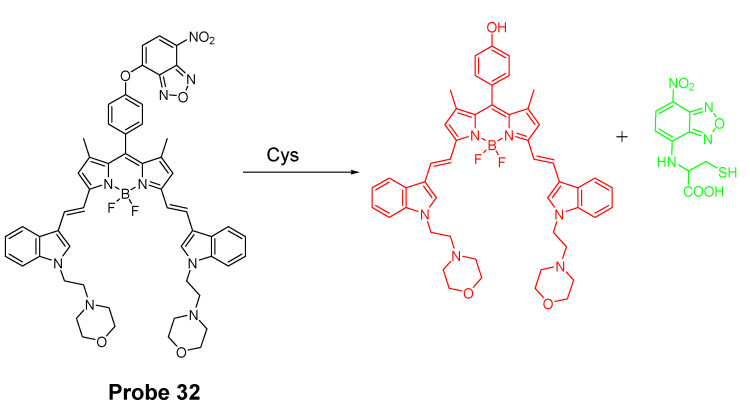
Structure of **Probe 32** and its reaction with Cys.

**Figure 32 molecules-26-03575-f032:**
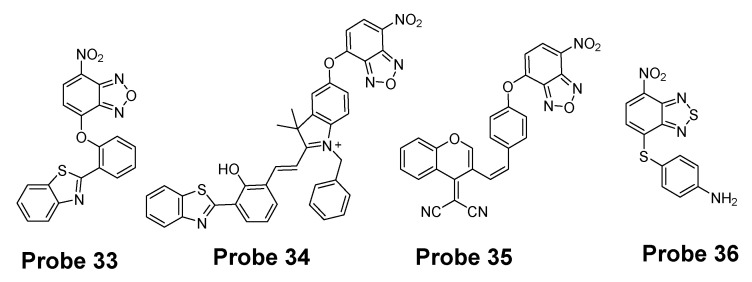
Structures of **Probe 33**, **34**, **35**, and **36**.

**Figure 33 molecules-26-03575-f033:**
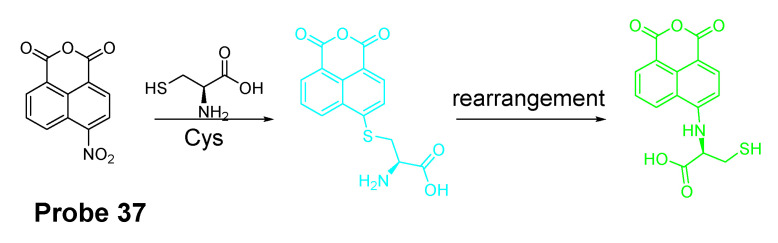
Structure of **Probe 37** and its reaction with Cys.

**Figure 34 molecules-26-03575-f034:**
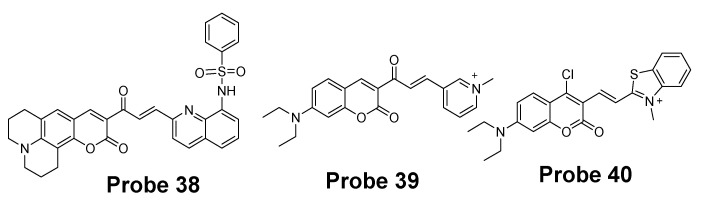
Structures of **Probes 38**, **39**, and **40**.

**Figure 35 molecules-26-03575-f035:**
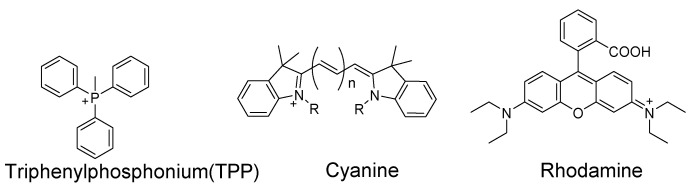
Representative mitochondrion-targeting.

**Figure 36 molecules-26-03575-f036:**
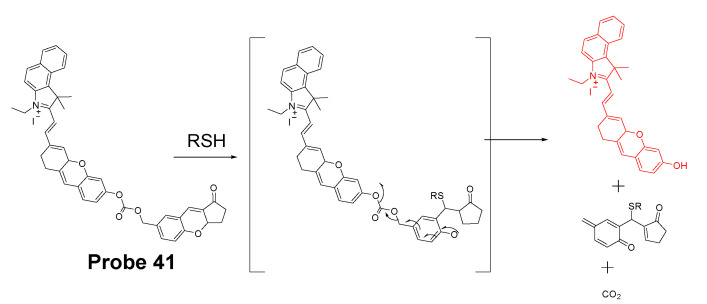
Structure of **Probe 41** and its reaction with thiols.

**Figure 37 molecules-26-03575-f037:**
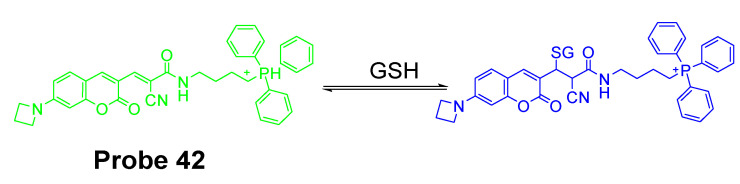
Structure of **Probe 42** and its reversible reaction with GSH.

**Figure 38 molecules-26-03575-f038:**
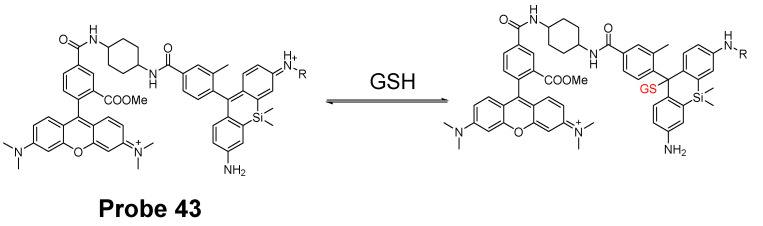
Structure of **Probe 43** and its reversible reaction with GSH.

**Figure 39 molecules-26-03575-f039:**
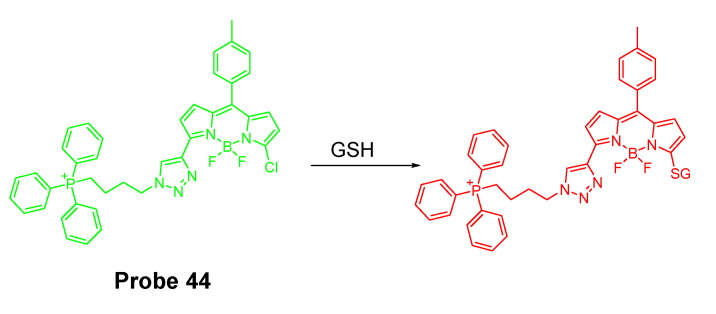
**Probe 44** and its reaction with GSH.

**Figure 40 molecules-26-03575-f040:**
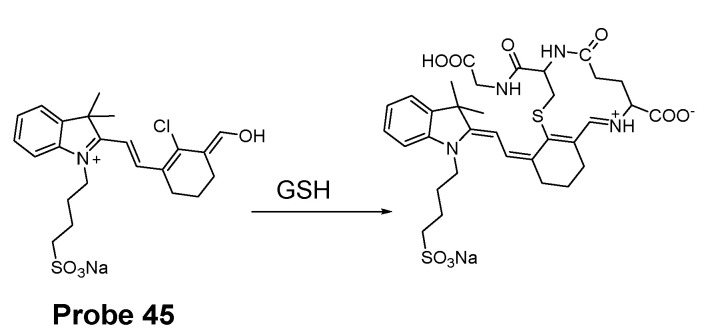
Structure of **Probe 45** and its reaction with GSH.

**Figure 41 molecules-26-03575-f041:**
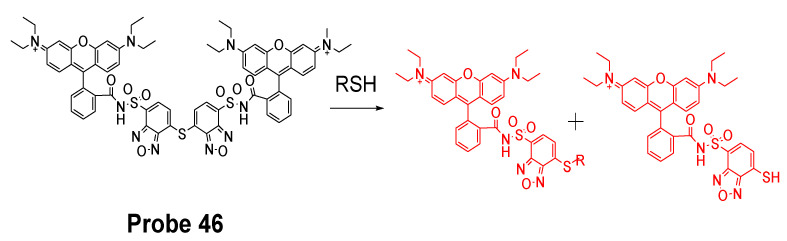
Structure of **Probe 46** and its reaction with thiols.

**Figure 42 molecules-26-03575-f042:**
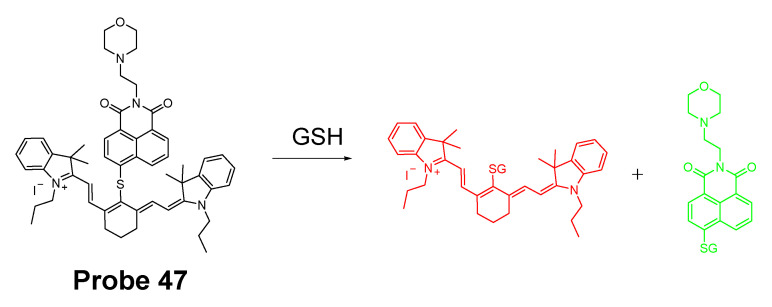
Structure of **Probe 47** and its reaction with GSH.

**Figure 43 molecules-26-03575-f043:**
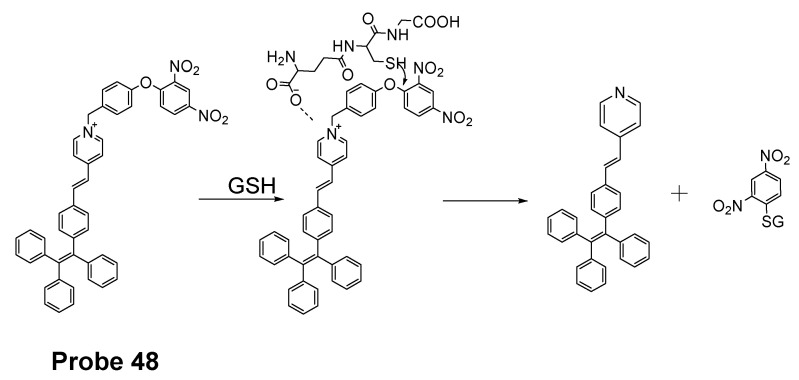
Structure of **Probe 48** and its reaction with GSH.

**Figure 44 molecules-26-03575-f044:**
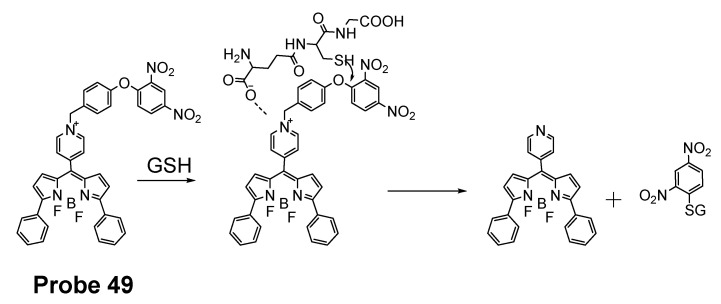
Structure of **Probe 49** and its reaction with GSH.

**Figure 45 molecules-26-03575-f045:**
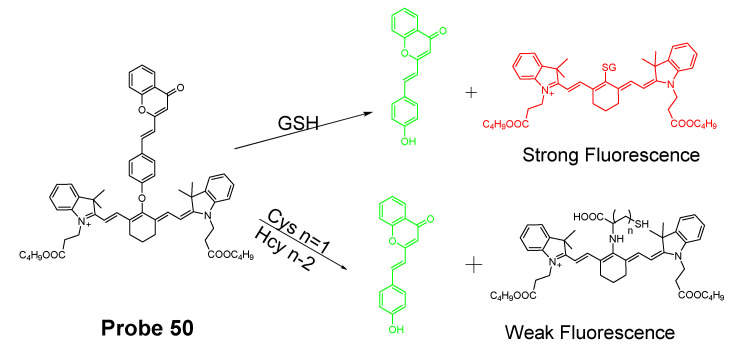
Structure of **Probe 50** and its reaction with GSH and Cys/Hcy.

**Figure 46 molecules-26-03575-f046:**
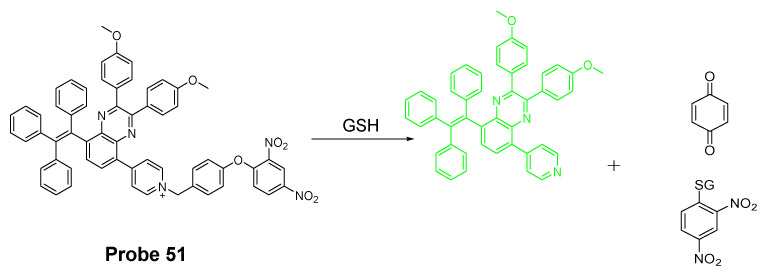
Structure of **Probe 51** and its reaction with GSH.

**Figure 47 molecules-26-03575-f047:**
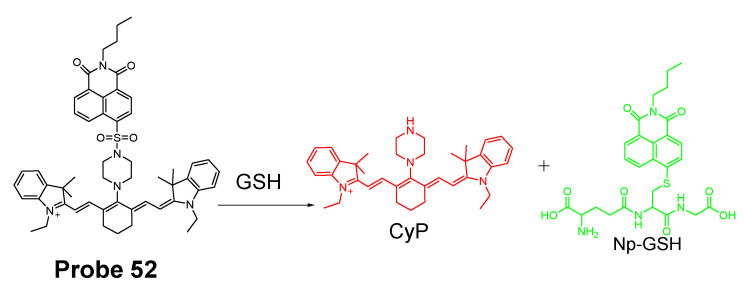
Structure of **Probe 52** and its reactions with thiols.

**Figure 48 molecules-26-03575-f048:**
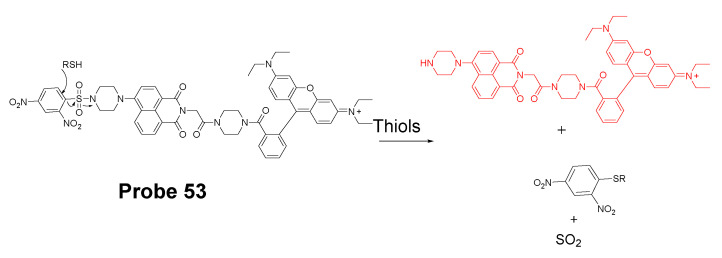
Structure of **Probe 53** and its reaction with thiols.

**Figure 49 molecules-26-03575-f049:**
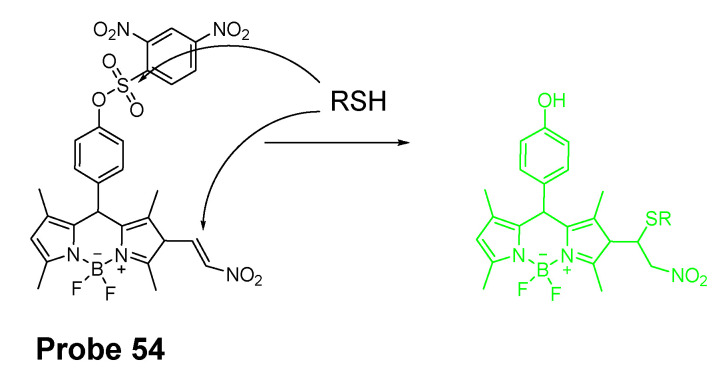
Structure of **Probe 54** and its reaction with thiols.

**Figure 50 molecules-26-03575-f050:**
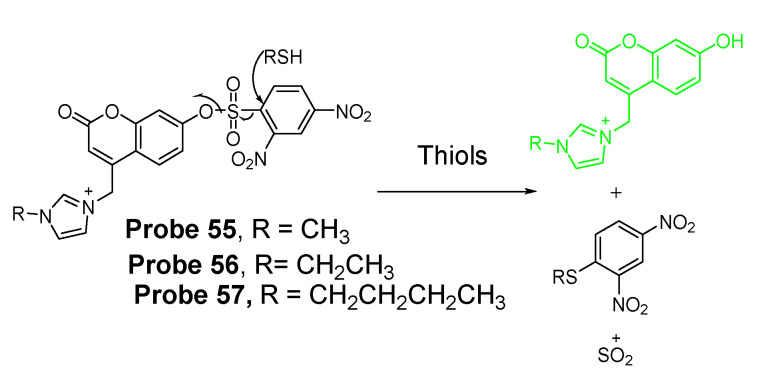
Structures of **Probes 55**–**57** and their reactions wih thiols.

**Figure 51 molecules-26-03575-f051:**
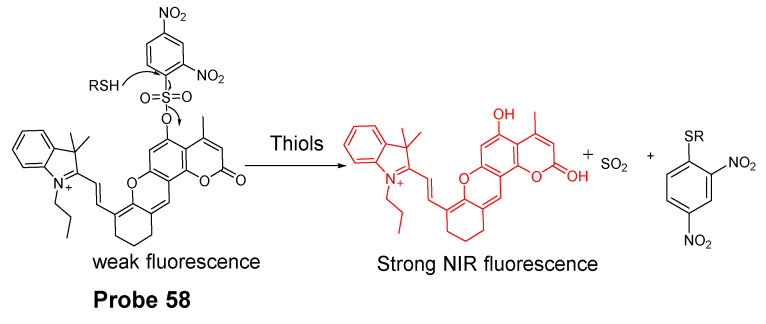
Structure of **Probe 58** and its reaction with thiols.

**Figure 52 molecules-26-03575-f052:**
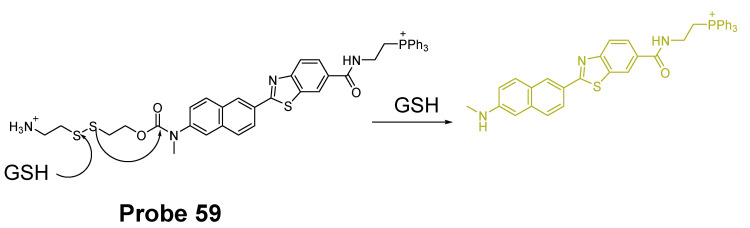
Structure of **Probe 59** and its reaction with GSH.

**Figure 53 molecules-26-03575-f053:**
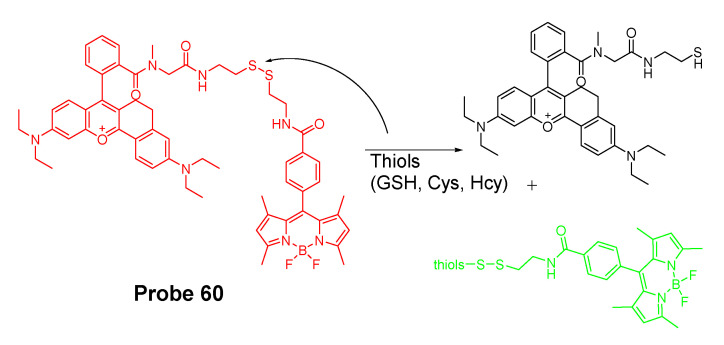
Structure of **Probe 60** and its reaction with thiols.

**Figure 54 molecules-26-03575-f054:**
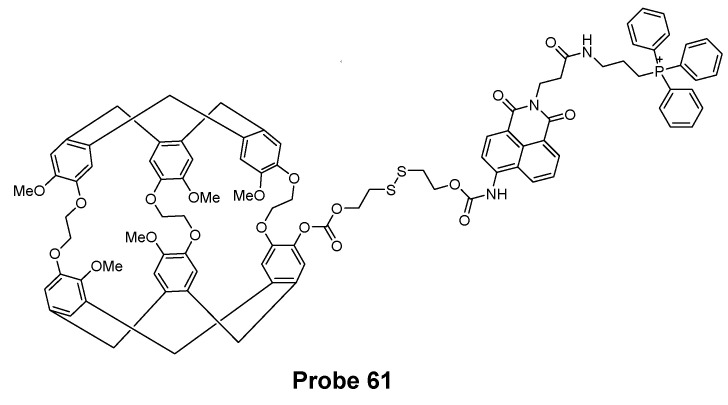
Structure of **Probe 61**.

**Figure 55 molecules-26-03575-f055:**
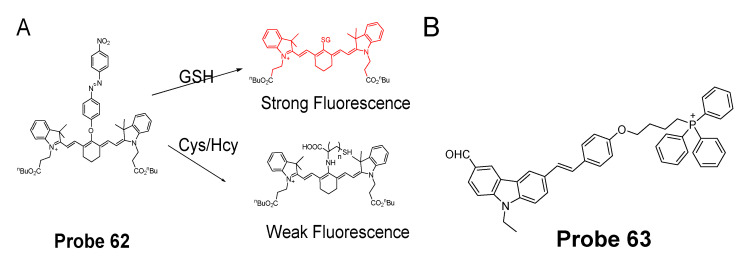
(**A**). Structure of **Probe 62** and its reaction with GSH and Cys/Hcy; (**B**). Structure of **Probe 63**.

**Figure 56 molecules-26-03575-f056:**
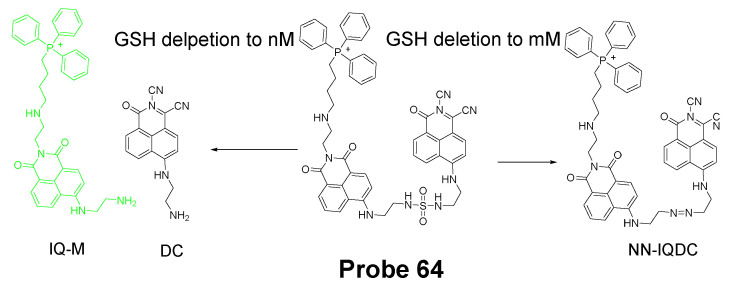
**Probe 64** and its reaction with different levels of GSH.

**Figure 57 molecules-26-03575-f057:**
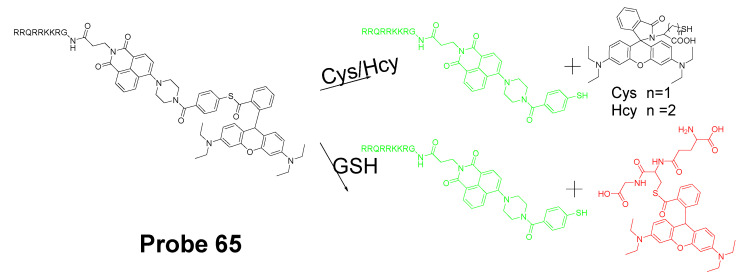
**Probe 65** and its reaction with GSH and Cys/Hcy.

**Figure 58 molecules-26-03575-f058:**
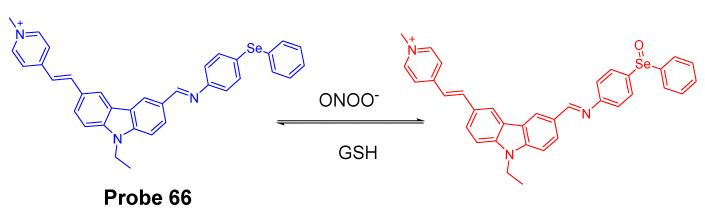
Structure of **Probe 66** and its reaction with ONOO^−^/GSH.

**Figure 59 molecules-26-03575-f059:**
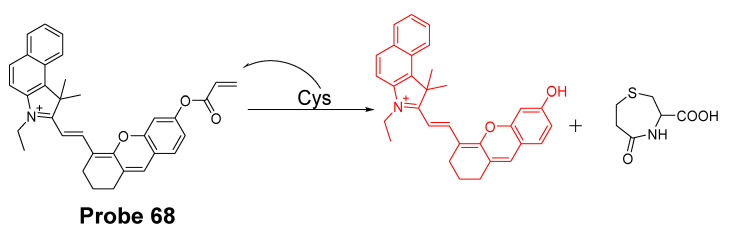
Structure of **Probe 68** and its reaction with Cys.

**Figure 60 molecules-26-03575-f060:**
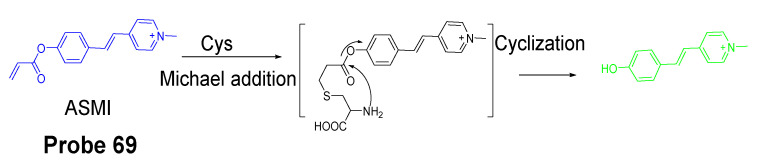
Structure of **Probe 69** and its reaction with Cys.

**Figure 61 molecules-26-03575-f061:**
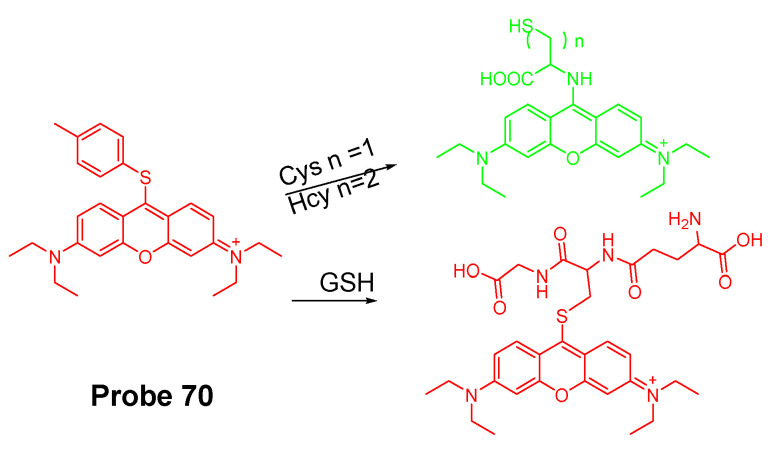
Structure of **Probe 70** and its reaction with GSH and Cys/Hcy.

**Figure 62 molecules-26-03575-f062:**
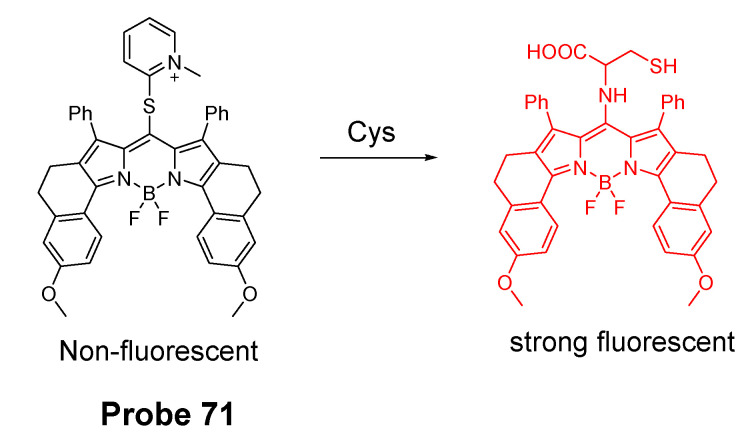
Structure of **Probe 71** and its reaction with Cys.

**Figure 63 molecules-26-03575-f063:**
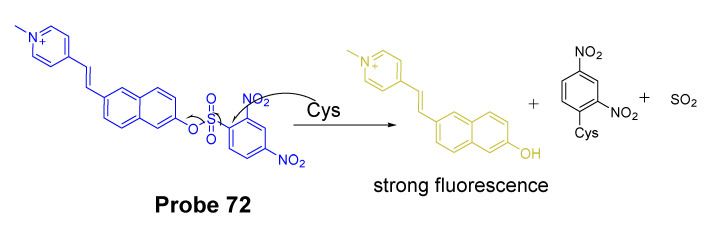
Structure of **Probe 72** and its reaction with Cys.

**Figure 64 molecules-26-03575-f064:**
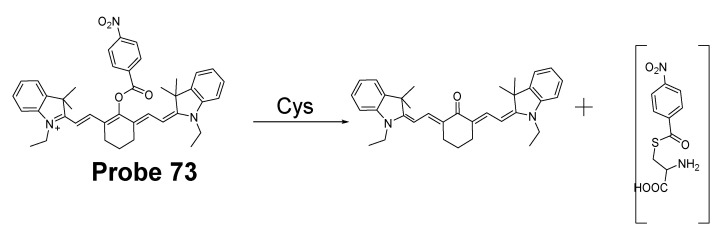
Structure of **Probe 73** and its reaction with Cys.

**Figure 65 molecules-26-03575-f065:**
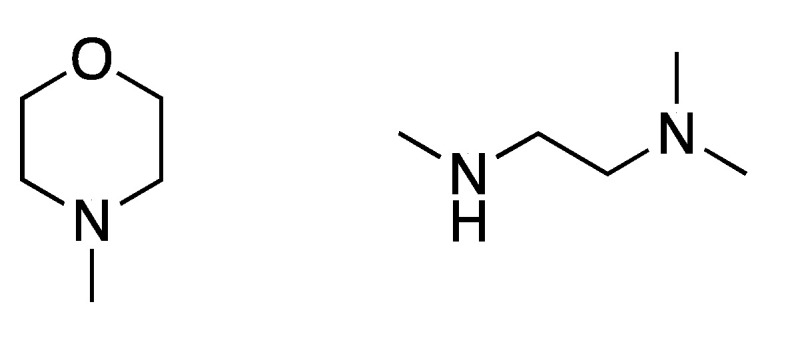
Representative lysosome-targeting.

**Figure 66 molecules-26-03575-f066:**
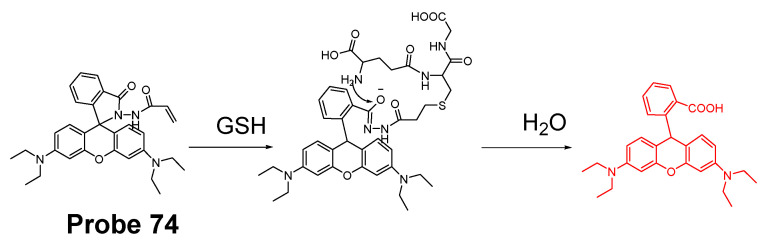
Structure of **Probe 74** and its reaction with GSH.

**Figure 67 molecules-26-03575-f067:**
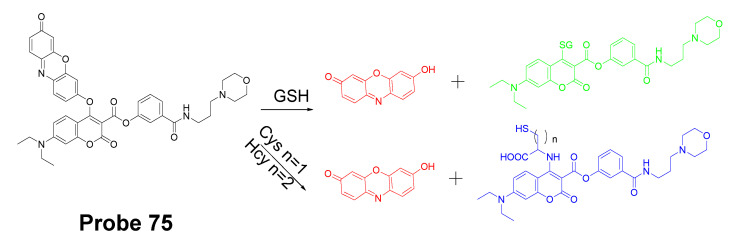
Structure of **Probe 75** and its reaction with GSH and Cys/Hcy.

**Figure 68 molecules-26-03575-f068:**
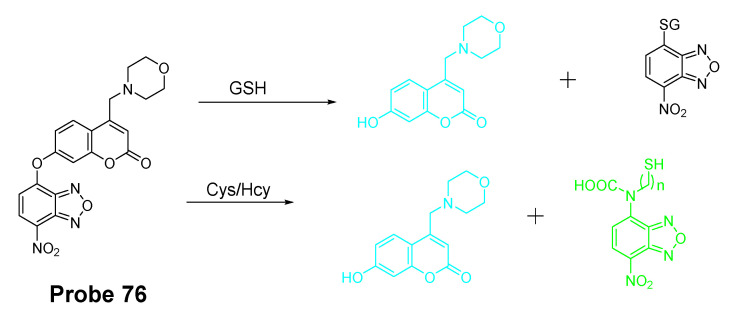
Structure of **Probe 76** and its reactions with GSH or Cys/Hcy.

**Figure 69 molecules-26-03575-f069:**
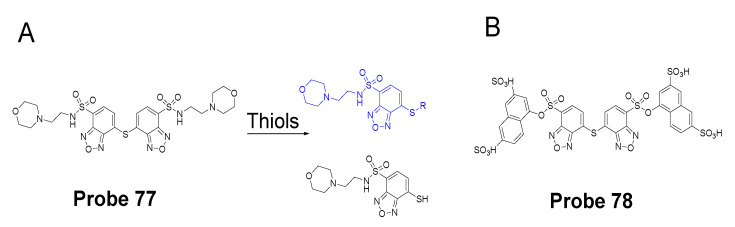
A. Structure of **Probe 77** and its reaction with thiols; B. Structure of **Probe 78**.

**Figure 70 molecules-26-03575-f070:**
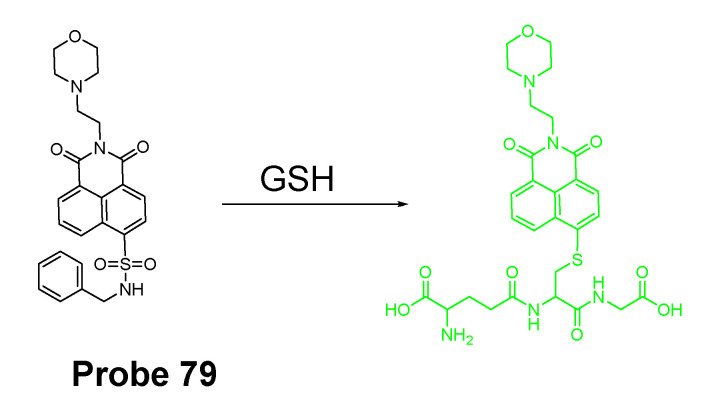
Structure of **Probe 79** and its reaction with GSH.

**Figure 71 molecules-26-03575-f071:**
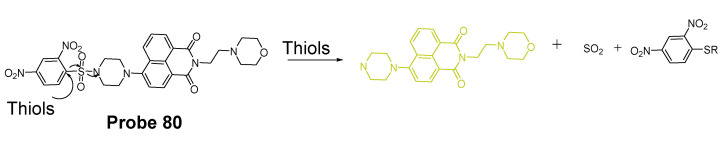
Structure of **Probe 80** and its reaction with thiols.

**Figure 72 molecules-26-03575-f072:**
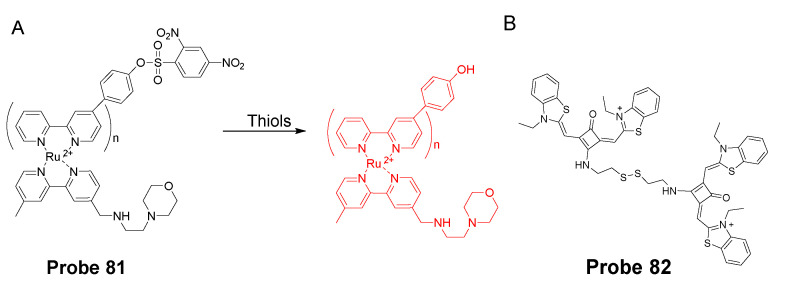
(**A**). Structure of **Probe 81** and its reaction with thiols. (**B**). Structure of **Probe 82**.

**Figure 73 molecules-26-03575-f073:**
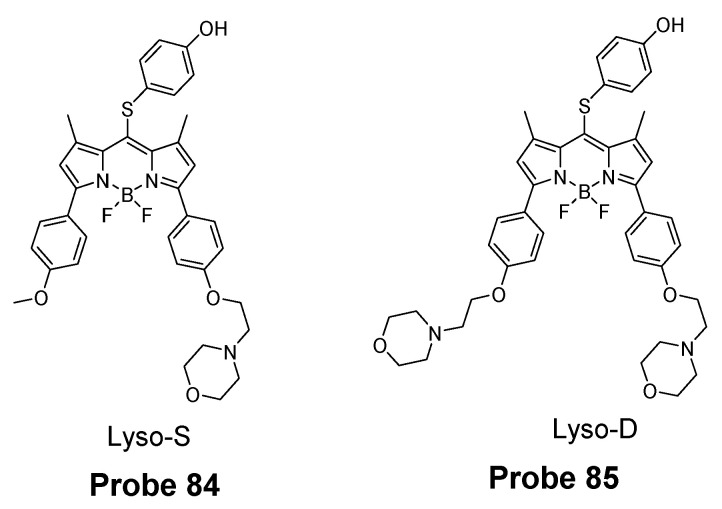
Structures of **Probes 84**, and **85**.

**Figure 74 molecules-26-03575-f074:**
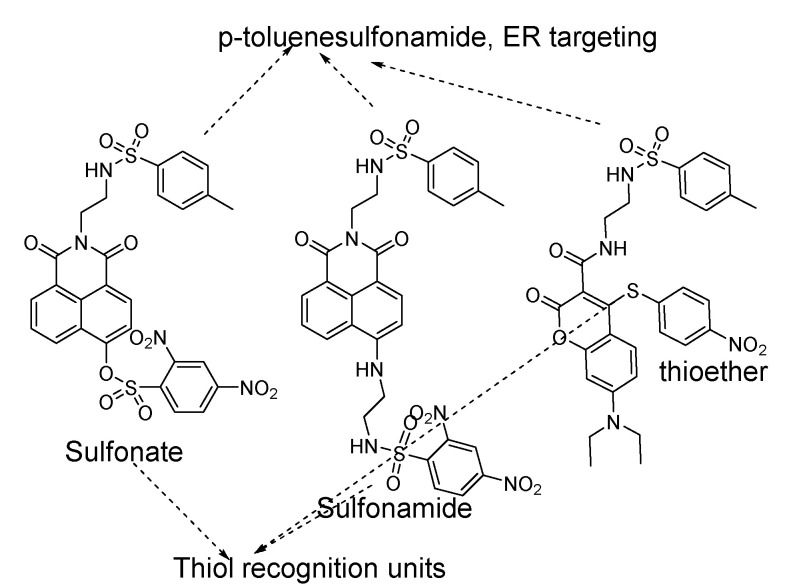
ER-targeting thiols imaging probes.

**Figure 75 molecules-26-03575-f075:**
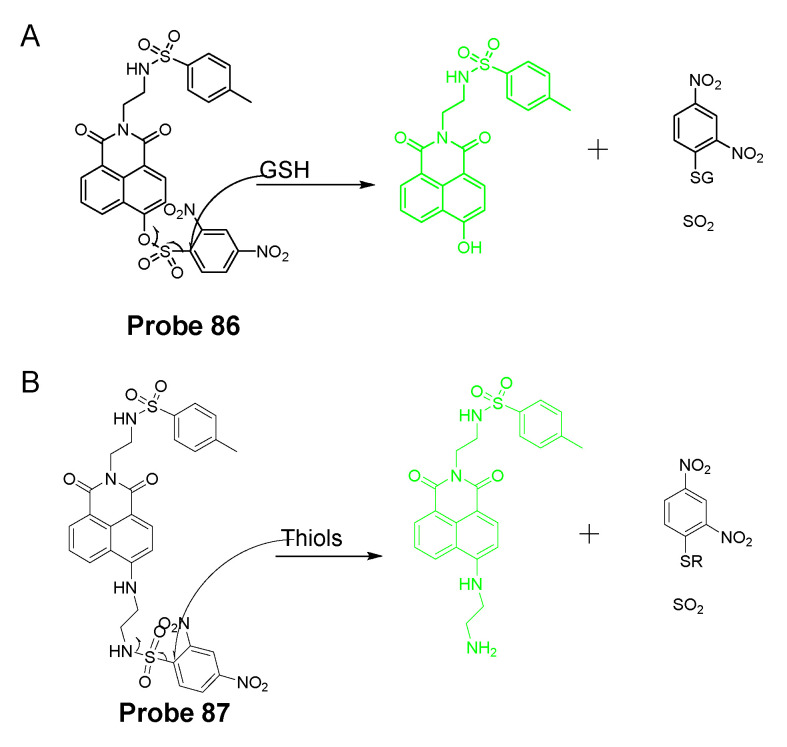
(**A**). Structure of **Probe 86** and its reaction with GSH; (**B**). Structure of **Probe 87** and its reactions with thiols.

**Figure 76 molecules-26-03575-f076:**
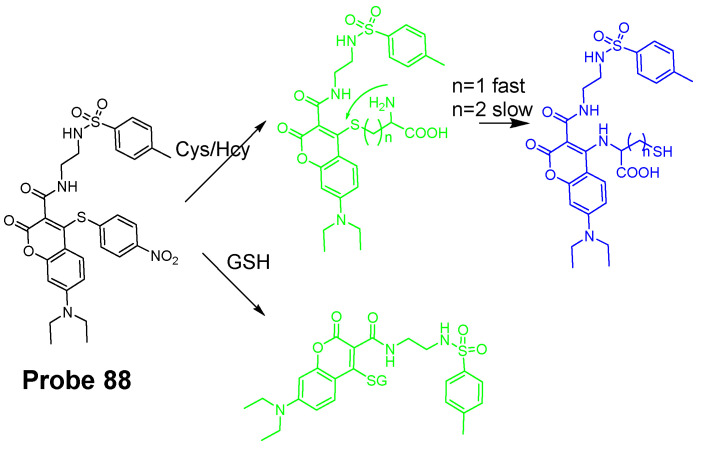
Structure of **Probe 88** and its different reactions mechanisms toward GSH or Cys/Hcy.

**Table 1 molecules-26-03575-t001:** Thiol imaging probes presented in the review.

Probe Number	Mechanism	Usage	Detection Limit	Reference Number
**Probe 1**	Michael addition	Whole cells	GSH (53 nM); Cys (<50 nM); Hcy (~100 nM)	34
**Probe 2**	Michael addition	Whole cells	NA	36
**Probe 3**	Michael addition	Whole cells	0.219 μM	71
**Probe 4**	Michael addition	Whole cells	GSH (0.085 μM); Cys (0.13 μM); Hcy (0.12 μM)	72
**Probe 5**	Michael addition	Whole cells	NA	73
**Probe 6**	Michael addition	Whole cells	NA	74
**Probe 7**	SNAr reaction	Whole cells	GSH (8.6 × 10^−8^ M)	29
**Probe 8**	SNAr reaction		GSH (0.07 μM) Cys (0.13 μM)	80
**Probe 9**	SNAr reaction	Whole cells	GSH (0.29 μM)	81
**Probe 10**	Thiol–sulfide exchange reaction	Whole cells	NA	42
**Probe 11**	SNAr reaction	Whole cells	NA	83
**Probes 12, 13**	SNAr reaction	Whole cells	GSH (84 nM)	84
**Probe 14**	cleavage of sulfonamide	Whole cells	NA	85
**Probe 15**	cleavage of sulfonamide	Whole cells	NA	88
**Probe 16**	cleavage of sulfonamide	Whole cells	GSH (1.9 μM)	89
**Probe 17**	cleavage of sulfonate ester	Whole cells	GSH (1.8 × 10^−8^ M)	61
**Probe 18**	cleavage of sulfonate ester	Whole cells	NA	91
**Probe 19**	cleavage of sulfonate ester	Whole cells	GSH (0.17 μM)	67
**Probe 20**	cleavage of a disulfide bond	Whole cells	GSH (28 μM)	92
**Probe 21**	cleavage of a disulfide bond	Whole cells	NA	93
**Probe 22**	cleavage of a disulfide bond	Whole cells	GSH (0.86 × 10^−6^ M)	94
**Probe 23**	cyclization of Cys/Hcy with acrylates or aldehydes	Whole cells (Cys selective)	NA	101
**Probe 24**	cyclization of Cys/Hcy with acrylates or aldehydes	Whole cells (Cys selective)	Cys (0.19 μM)	38
**Probe 25**	cyclization of Cys/Hcy with acrylates or aldehydes	Whole cells (Cys selective)	Cys (∼0.2 μM)	44
**Probe 26**	cyclization of Cys/Hcy with bromoacetylfluorescein monoaldehyde	Whole cells (Cys selective)	Cys (0.51 μM)	114
**Probes 27, 28**	cyclization	Whole cells (Hcy selective)	Hcy (P-Hcy-1, 1.94 × 10^−6^ M; P-Hcy-2, 1.44 × 10^−7^ M)	43
**Probe 29**	Cys-induced SNAr substitution−rearrangement reaction	Whole cells (Cys selective)	Cys (2.12 × 10^−7^ M)	117
**Probe 30**	Cys-induced SNAr substitution−rearrangement reaction	Whole cells (Cys selective)	Cys (5.52 × 10^−7^ M)	118
**Probe 31**	Cys-induced SNAr substitution−rearrangement reaction	Whole cells (Cys/Hcy selective)	Cys (2.00 × 10^−8^ M) Hcy (1.02 × 10^−8^ M)	120
**Probe 32**	Cys-induced SNAr substitution−rearrangement reaction	Whole cells (Cys selective)	Cys (22 nM)	121
**Probe 33**	Cys-induced SNAr substitution−rearrangement reaction	Whole cells, discrimination of Cys and GSH	Cys (0.08 μM) GSH (0.06 μM)	124
**Probe 34**	Cys-induced SNAr substitution−rearrangement reaction	Whole cells, discrimination of Cys/Hcy, GSH, and H_2_S	GSH (4.30 μM) Cys (4.25 μM) Hcy (5.11 μM) H2S (6.74 μM)	125
**Probe 35**	Cys-induced SNAr substitution−rearrangement reaction	Whole cells, discriminate Cys/Hcy from GSH	Cys (2.1 × 10^−8^ M) Hcy 1.7 × 10^−8^ M; GSH (2.6 ×10^−8^ M)	126
**Probe 36**	Cys-induced SNAr substitution−rearrangement reaction	Whole cells, (Cys/Hcy selective)	Cys and Hcy (0.1 μM)	127
Probe 37	Cys-induced SNAr substitution−rearrangement reaction	Whole cells, (Cys selectvie)	Cys (0.3 μM)	128
**Probe 38**	Michael addition reaction in combination with a steric hinderance factor.	Whole cells (Cys selective)	Cys (10^−7^ M)	129
**Probe 39**	Michael addition assisted by an electrostatic attraction.	Whole cells (Cys selective)	Cys (25 nM)	130
**Probe 40**	SNAr substitution−rearrangement reaction	Whole cells simultaneously detects Cys and GSH	Cys (>0.4 μM) GSH (0.05 μM)	102
**Probe 41**	Michael addition followed by self-immolative reaction	Mitochondrial thiols	GSH (0.59 μM) Cys (0.39 μM) Hcy (0.54 μM)	137
**Probe 42**	Michael addition	Mitochondrial GSH	NA	138
**Probe 43**	Michael addition	Mitochondrial GSH	NA	139
**Probe 44**	SNAr reaction	Mitochondrial GSH	GSH (1.1 μM)	140
**Probe 45**	SNAr reaction	Mitochondrial GSH	GSH (24.16 μM)	141
**Probe 46**	Thiol–sulfide exchange reaction	Mitochondrial thiols	NA	22
**Probe 47**	SNAr reaction	Mitochondrial and lysosomal GSH	GSH (1 nM)	143
**Probe 48**	SNAr reaction	Mitochondrial thiols	GSH (0.61 μM).	35
**Probe 49**	SNAr reaction	Mitochondrial GSH	GSH (109 nM)	160
**Probe 50**	SNAr reaction	Mitochondrial GSH	GSH (24 nM, visible) and (32 nM, NIR)	1
**Probe 51**	Cleavage of the dinitrophenyl ether	Mitochondrial GSH	GSH (434 nM)	161
**Probe 52**	cleavage of sulfonamide	Mitochondrial GSH	GSH (1.53 × 10^−7^ M, visible channel) and (1.71 × 10^−7^ M, NIR channel)	162
**Probe 53**	cleavage of sulfonamide	Mitochondrial thiols	GSH (0.89 μM); Cys (0.47 μM), Hcy (2.4 μM)	164
**Probe 54**	cleavage of sulfonate ester	Mitochondrial thiols	Hcy (87 nM) Cys (147 nM) GSH (129 nM)	167
**Probes 55, 56, 57**	cleavage of sulfonate ester	Mitochondrial thiols	GSH (**Probe 55**, 31.4 nM; **Probe 56**, 29.2 nM; **Probe 57**, 29.6 nM)	171
**Probe 58**	cleavage of sulfonate ester	Mitochondrial thiols	GSH (0.11μM) Cys (0.08 μM) Hcy (0.20 μM)	172
**Probe 59**	cleavage of disulfide bond	Mitochondrial thiols	NA	173
**Probe 60**	cleavage of disulfide bond	Mitochondrial thiols	GSH (0.26 μM)	174
**Probe 61**	cleavage of disulfide bond	Mitochondrial thiols	GSH (10^−10^ M, using Hyper-CEST NMR)	178
**Probe 62**	1, 6-conjugate addition and subsequent elimination reaction	Mitochondrial GSH	GSH (26 nM)	179
**Probe 63**	Others	Mitochondrial thiols	Cys (0.2 μM)	180
**Probe 64**	cleavage of sulfonamide for ultratrace change of GSH	Mitochondrial GSH	GSH (2.02 nM)	181
**Probe 65**	Others	Mitochondrial GSH	GSH (5.15 μM) Cys (0.865 μM) Hcy (6.51 μM)	135
**Probe 66**	Others	Mitochondrial ONOO^−^/GSH levels	NA	186
**Probe 67**	Others	Mitochondrial redox potential	NA	187
**Probe 68**	cyclization of Cys with acrylates or aldehydes	Mitochondrial Cys	Cys (14.5 nM)	188
**Probe 69**	cyclization of Cys with acrylates or aldehydes	Mitochondrial Cys	NA	112
**Probe 70**	Cys-induced SNAr substitution−rearrangement reaction	Mitochondrial Cys/Hcy	Cys (22 nM) Hcy (23 nM)	192
**Probe 71**	SNAr substitution−rearrangement reaction	Mitochondrial Cys	Cys (72 nM)	193
**Probe 72**	cleavage of sulfonamide	Mitochondrial Cys	Cys (0.29 μM)	194
**Probe 73**	Others	Mitochondrial Cys	Cys (0.2 μM)	195
**Probe 74**	Michael addition	Lysosomal GSH	GSH (190 nM)	203
**Probe 75**	SNAr reactions	Lysosomal thiols	GSH (16 nM) Cys (27 nM) Hcy (33 nM)	204
**Probe 76**	SNAr reactions	Lysosomal thiols	GSH (3.9 × 10^−8^ M); Cys (3.3 × 10^−8^ M); Hcy (5.2 × 10 ^−8^ M)	205
**Probe 77**	Thiol–sulfide exchange reaction	Lysosomal thiols	NA	206
**Probe 78**	Thiol–sulfide exchange reaction	Lysosomal thiols	NA	207
**Probe 79**	cleavage of sulfonamide	Lysosomal GSH	NA	208
**Probe 80**	cleavage of sulfonamide	Lysosomal thiols	GSH (2.41 × 10^−6^ M); Cys (2.6 × 10^−7^ M); Hcy (4.87 × 10^−6^ M)	165
**Probe 81**	cleavage of sulfonate	Lysosomal thiols	GSH (62 nM); Cys (146 nM); Hcy (115 nM)	209
**Probe 82**	cleavage of disulfide	Lysosomal GSH	GSH (0.15 μM)	210
**Probe 83**	others	Lysosomal GSH	GSH (1.03 μM)	211
**Probes 84, 85**	Cys-induced SNAr substitution followed by a intramolecular rearrangement	Lysosomal Cys	Cys (46 nM for Lyso-S and 76 nM for Lyso-D)	212
**Probe 86**	cleavage of sulfonate	Endoplasmic reticulum GSH	NA	215
**Probe 87**	cleavage of sulfonamide	Endoplasmic reticulum thiols	GSH (4.70 × 10^−6^ M); Cys (1.67 × 10^−7^ M); Hcy (9.62 × 10^−7^ M)	214
**Probe 88**	SNAr substitution	Endoplasmic reticulum thiols	GSH (23 nM), Cys (14 nM), Hcy (16 nM)	216

## Data Availability

Not applicable.

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
