# Peer review of "Fluorescent Probes for Live Cell Thiol Detection"

_molecules, 2021, doi:10.3390/molecules26123575_

Round 1

Reviewer 1 Report

In the review, the methods for fluorescent identification of the presence of various thioles in living cells and their organelles are summarized. The progress of the last two decades is mostly considered. The mechanisms of fluorescent molecules formation due to the use of various reagents, the peculiarities of fluorescence response as wel as practical applications of the methods are discussed.
The review is informative and well written. I suppose, the review can be published almost in its present form except for some necessary clarifications.

The mechanisms shown in Figures 17, 18, 19 represent just hydrolyses of the 2,4-dinitrophenylsulfonates. How the thiols are involved in them? The same applies to Figures 36, 48, 50, 51, 56, 59, 63, 75 where the reactions represent similar hydrolytic (?) bonds cleavage and the role of the thiols is unclear. Or the reactions with 2,4-dinitrophenylsulfonates lead to release of SO2 as it is shown in Fig. 71?

Figure 46 (probe 51). The mechanism is described in the corresponding text as "Cleavage of the dinitrophenyl ether", however, in the Figure the dequaternization of pyridinium salt is shown.

I have also noticed some grammar flaws:

Page 4: More importantly, the probe is also the first reAgent being used to monitor the thiols status in zebrafish...

Paga 7: A fluorescence emission wavelength red shift (556 nm to ~588 nm) occurred after the reaction, and the ratio of fluorescence intensity (I588/I556) linearly increased as the increasing concentration of GSH (0−60 μM) WAS INCREASING, with the detection limit...

Page 18: The probe itself shows barEley any fluorescence...

Page 21: While most OF the probes being developed...

Page 27: In 2016, Jian Zhang et al. developED a probe...

Page 27: A linear relationship were WAS observed between the fluorescence intensity...

Page 36: This destroys the photoinduced electron transfer (PET) process and restoreS the strong NIR fluorescence...

Page 44: Fluorescence quenching is achieved viA a fluorescence resonance energy transfer...

Page 47: Most of the probes developed are based on an irreversible reAction with thiols...

Also, it is unclear why some of the lines in the last paragraph before the Conclusion are bold ("ER-CP composed of three parts", etc.).

That's all.

Author Response

Dear Reviewer 1:

Thank you for your time and efforts in reviewing the manuscript. Your comments help improve the quality of the manuscript. We really appreciate it. Accordingly, we have addressed all your comments (see below).

Point-by-point Response to Reviewers' Comments

Reviewer 1.

Comments: The mechanisms shown in Figures 17, 18, 19 represent just hydrolyses of the 2,4-dinitrophenylsulfonates. How the thiols are involved in them? The same applies to Figures 36, 48, 50, 51, 56, 59, 63, 75 where the reactions represent similar hydrolytic (?) bonds cleavage and the role of the thiols is unclear. Or the reactions with 2,4-dinitrophenylsulfonates lead to release of SO2 as it is shown in Fig. 71?

Response: The authors thank the reviewer for the comment. Although the products were the same as that from the hydrolysis in those reactions, they were generated from the thiol-initiated reactions not from hydrolysis. To avoid confusion, the reaction schemes have been modified to reflect the thiol-initiated reactions in Figures 36, 48, 50, 51, 59, 63 and 75. For Figure 56, a sentence “from the product IQ-M which was a result of a thiol attack at the sulfonamide bond (addition reaction) followed by an elimination reaction to release IQ-M.” was added in the paragraph to clarify the reaction.

The reaction scheme in Fig. 71 has been modified to reflect the release of SO2.

Comment: Figure 46 (probe 51). The mechanism is described in the corresponding text as "Cleavage of the dinitrophenyl ether", however, in the Figure the dequaternization of pyridinium salt is shown.

Response: The author would like to thank the reviewer for the comment. The sentence has been changed to “Cleavage of the dinitrophenyl ether followed by a self-immolation reaction” to reflect the process of dequaternization of pyridinium. Accordingly, the reaction scheme has been modified.

Comments: I have also noticed some grammar flaws:

Responses: All gramma errors raised by the reviewer have been corrected (see below)

Page 4: More importantly, the probe is also the first reAgent being used to monitor the thiols status in zebrafish...

Corrected

Paga 7: A fluorescence emission wavelength red shift (556 nm to ~588 nm) occurred after the reaction, and the ratio of fluorescence intensity (I588/I556) linearly increased as the increasing concentration of GSH (0−60 μM) WAS INCREASING, with the detection limit...

Corrected.

Page 18: The probe itself shows barEley any fluorescence...

BarEley has been corrected (barely)

Page 21: While most OF the probes being developed...

Corrected (page 22)

Page 27: In 2016, Jian Zhang et al. developED a probe...

Corrected

Page 27: A linear relationship were WAS observed between the fluorescence intensity...

Corrected

Page 36: This destroys the photoinduced electron transfer (PET) process and restoreS the strong NIR fluorescence...

Corrected

Page 44: Fluorescence quenching is achieved viA a fluorescence resonance energy transfer...

Corrected

Page 47: Most of the probes developed are based on an irreversible reAction with thiols...

Corrected

Also, it is unclear why some of the lines in the last paragraph before the Conclusion are bold ("ER-CP composed of three parts", etc.).

All corrected

Reviewer 2 Report

In this review, Wang et al. summarized the development of fluorescent probes for selectively detecting cellular and subcellular thiols. The organization of materials, discussion, and author’s perspectives on the significance, current, and future development of fluorescent probes for thiols in living cells are great. The reviewer recommends it for publication in this journal after revision.

1) I was just curious that if there is any probe for PSH specifically. I understand probes for those small molecules (GSH for example) are possibly enough to detect PSH. But I am still wondering if selectively detecting PSH has any merits/significance or not?

2) The text above Figure 4 has some Font size error

3) I was just wondering if authors can add more introduction/comparison about the different mechanisms of those probes. Are there any specific merits for each mechanism in detecting thiols?

4) For developing an efficient probe, what are the requirements for detecting thiol in cell or organelle respectively?

5) I think there is a lack of the author’s perspective on the development of those probes. There are only some simple mentions of the probe application throughout the whole review. But I suggest that those applications/significances/implementations should be more thoroughly illustrated. Adding more figures is also helpful.

6) I highly suggested that adding some Tables to summarize this review, for example probes (number), name, mechanism, target, for the whole cell or organelles, detection limit, application, reference number etc..

Author Response

Dear Reviewer 2:
Thank you for your time and efforts in reviewing the manuscript. Your comments help improve the quality of the manuscript. We really appreciate it. Accordingly, we have addressed all your comments (see below).

Point-by-point Response to Reviewers' Comments

Reviewer 2

Comment 1: I was just curious that if there is any probe for PSH specifically. I understand probes for those small molecules (GSH for example) are possibly enough to detect PSH. But I am still wondering if selectively detecting PSH has any merits/significance or not?

Response: The reviewer raised a valid issue. Yes, it will be valuable to have an agent that can selectively detect PSH. Unfortunately, this is most likely not possible since if a reagent can react with a PSH, it will react with NPSH since it is easier to react with NPSH. That is the reason we saw quite a few reagents presented in the review was able to react with NPSH not PSH since accessing thiols in proteins will run into an issue of steric hindrance which will slow or even prevent the reaction.

Comment 2: The text above Figure 4 has some Font size error

Response: Corrected

Comments 3: I was just wondering if authors can add more introduction/comparison about the different mechanisms of those probes. Are there any specific merits for each mechanism in detecting thiols?

Response: The authors would like to thank the reviewer for the comment. We have added a few paragraphs (e.g., paragraph 3 in section 2.1 and a paragraph in conclusion) in the review to address this issue.

Comment 4: For developing an efficient probe, what are the requirements for detecting thiol in cell or organelle respectively?

Response: A paragraph was added in page 3 to address this issue.

Comment 5: I think there is a lack of the author’s perspective on the development of those probes. There are only some simple mentions of the probe application throughout the whole review. But I suggest that those applications/significances/implementations should be more thoroughly illustrated. Adding more figures is also helpful.

Response: The authors would like to thank the reviewer for the suggestion. We have added our perspectives wherever appropriate in the review, such as paragraph 3 in section 2.1 and a paragraph in conclusion.

Comment 6: I highly suggested that adding some Tables to summarize this review, for example probes (number), name, mechanism, target, for the whole cell or organelles, detection limit, application, reference number etc..

Response: The authors would like to thank the reviewer for the suggestion. We included a table in the conclusion to summarize the probes, their mechanisms of thiol detection, usage, detection limits, and references.